# D-CORE: Incentivizing Task Decomposition in Large Reasoning Models for Complex Tool Use

Bowen Xu [* 1]   Shaoyu Wu [* 1]   Hao Jiang [1]   Kai Liu [1]   Xin Chen [1]   Lulu Hu [1]   Bin Yang [1]

## Abstract

Effective tool use and reasoning are essential capabilities for large reasoning models (LRMs) to address complex real-world problems. Through empirical analysis, we identify that current LRMs lack the capability of sub-task decomposition in complex tool use scenarios, leading to Lazy Reasoning. To address this, we propose a two-stage training framework D-CORE (**D**ecomposing tasks and **Co**mposing **Re**asoning processes) that first incentivize the LRMs' task decomposition reasoning capability via self-distillation, followed by diversity-aware reinforcement learning (RL) to restore LRMs' reflective reasoning capability. D-CORE achieves robust tool-use improvements across diverse benchmarks and model scales. Experiments on BFCLv3 demonstrate superiority of our method: D-CORE-8B reaches 77.7% accuracy, surpassing the best-performing 8B model by 5.7%. Meanwhile, D-CORE-14B establishes a new state-of-the-art at 79.3%. Code, model, and data are available at `https://github.com/alibaba/EfficientAI`.

## 1. Introduction

Tool use equips Large Language Models (LLMs) with the ability to invoke external interfaces, serving as a cornerstone for autonomous agents (Jimenez et al., 2024; Wei et al., 2025; Xie et al., 2024; Zhou et al., 2023). As tasks evolve from simple queries to compositional workflows (Qiao et al., 2024; Shen et al., 2024), recent benchmarks underscore the necessity for robust reasoning in real-world scenarios (Yao et al., 2024; Patil et al.). However, current paradigms face a dichotomy. Conventional tool use LLM approaches dominated by rule-based SFT (Liu et al., 2024b;a; Zhong et al., 2025; Prabhakar et al., 2025; Yin et al., 2025; Chen et al.,

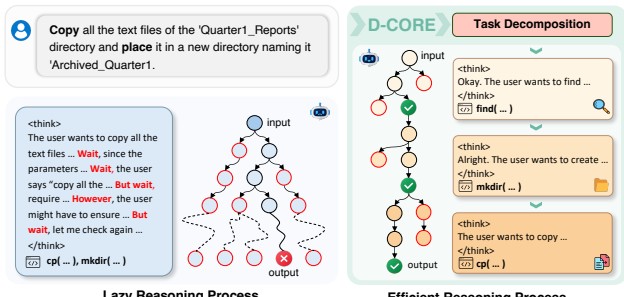

*Figure 1.* Comparison of baseline and D-CORE trained LRMs in complex tool use scenarios. Baseline LRMs exhibit "Lazy Reasoning" with repetitive reflection and incorrect answers, while D-CORE trained LRMs decompose tasks into executable subtasks.

2023), suffering from poor generalization in complex scenarios (Chen et al., 2025a; Chu et al., 2025). Conversely, while the RL-enhanced LRMs demonstrates success in math (Guo et al., 2025; Yang et al., 2025; Anthropic, 2025; OpenAI, 2024b; 2025) and single-turn tool use tasks (Qian et al., 2025; Zhang et al., 2025), we observe a diminishing return in complex tool use scenarios: LRMs consume substantiallly more tokens for reasoning yet yield marginal performance gains over LLMs. This points to a fundamental challenge: **how to effectively translate reasoning computation into complex tool proficiency for LRMs**.

We investigate Qwen3-series LRMs (Yang et al., 2025) and observe a critical issue: while effective in single-turn tool use scenarios, they suffer form "Lazy Reasoning" in complex multi-turn contexts. The models generate extensive but meaningless reasoning processes, impeding RL optimization (Yue et al., 2025; Gandhi et al., 2025; Ning et al., 2025). We attribute this degradation to the lack of task decomposition, verified by the effectiveness of decomposition-based prompting (Khot et al., 2022b; Zhou et al., 2022).

Motivated by this, we propose D-CORE (**D**ecomposing tasks and **Co**mposing **Re**asoning processes). This framework explicitly enforces decomposition and diversity via self-distillation and diversity-aware GRPO (DA-GRPO). As shown in Figure 1, D-CORE converts inefficient reasoning cycles into effective, step-by-step processes. We first employ self-distillation to bootstrap task decomposition, or-

---

[1]Alibaba Cloud Computing, Alibaba Group. Correspondence to: Bowen Xu <bowen.xbw@alibaba-inc.com>, Shaoyu Wu <wushaoyu.wsy@alibaba-inc.com>.

*Proceedings of the 43[rd] International Conference on Machine Learning*, Seoul, South Korea. PMLR 306, 2026. Copyright 2026 by the author(s).

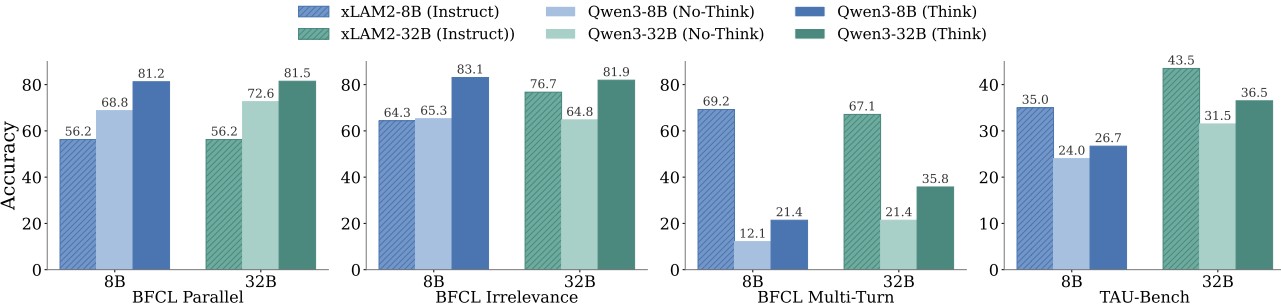

*Figure 2.* Comparison of LRM Qwen3 vs. instruct LLM xLAM2 performance on BFCLv3 Parallel, Irrelevance , Multi-turn and $\tau$-bench task.

ganizing sub-task executions into trajectories. However, this supervised approach tends to homogenize reasoning and reduce reflection. We remedy this via DA-GRPO, which adds an entropy regularization term to the advantage function. This design balances structural decomposition with reasoning diversity, enabling the LRM to autonomously decompose tasks and execute tools. Our main contributions are:

- We identify that LRMs lack the capability of sub-task decomposition in complex tool-use scenarios, leading to the phenomenon of "Lazy Reasoning".

- We develop a self-distillation framework that integrates task decomposition with reasoning process composition, enabling LRMs to acquire sophisticated sequential tool use strategies during reasoning without requiring additional human annotation.

- We propose DA-GRPO that incorporates entropy-based advantage functions to enable self-distillation LRMs to restore their reflection capabilities while maintaining task decomposition abilities, thereby addressing more complex tool use scenarios.

## 2. Tool Use Reasoning: Patterns and Limitations

### 2.1. Preliminary

**Tool use tasks.** Tool use tasks can be categorized into single-turn and multi-turn tasks based on context dependency. Single-turn tasks can be formulated as Markov decision process $M(P, T, Q) \to \tau$, where $P$ denotes the system policy, $T$ represents the available tool set, $Q$ corresponds to the current query, and $\tau$ represents tool call results. Upon decomposing query $Q$ into subtasks $S = \{s_1, s_2, \ldots, s_n\}$, three key scenarios for $M$ arise:

$$M \in \begin{cases} \text{Sequential: } s_i \text{ depends on output of } s_{i-1}, \\ \text{Parallel: } s_i \text{ can execute parrallely,} \\ \text{Irrelevant: } Q \text{ requires no tool use.} \end{cases}$$

The primary challenge lies in the fact that multi-intent and tool irrelevance $Q$ make $M$ particularly challenging for tool use LLMs (Liu et al., 2024a; Lin et al., 2024). Furthermore, multi-turn tool use scenarios (Yao et al., 2024; Patil et al.; Prabhakar et al., 2025) can be formulated as a $M(P, T, C, Q) \to \tau$, where $C$ denotes the conversation history. Unlike single-turn tasks, this formulation introduces additional complexity by requiring consideration of both the current query $Q$ intent and the long-term intent embedded in $C$.

**Reasoning process.** A reasoning process $\mathcal{R}$ refers to the sequence of intermediate steps through which a LRM arrives at its final answer. Within the LRM outputs examined in this paper, reasoning processes are specifically delimited by `<think>` and `</think>` tags. We controlled LRMs' reasoning processes through prompts containing `"<think>\n\n</think>\n\n"` tags to enable "no-think" modes. A thought $r$ is the basic logical block in a reasoning process. Given a reasoning process $\mathcal{R}$, it can be decomposed into an ordered sequence of thoughts: $\mathcal{R} = \{r_1, r_2, ..., r_n\}$, where $n$ represents the total number of thoughts.

### 2.2. Reasoning Process Enhances Tool Use Awareness

To examine the impact of reasoning on tool use, we evaluate on BFCLv3 (Patil et al.) (parallel, irrelevance, multi-turn) and $\tau$-bench (Yao et al., 2024). We benchmark the Qwen3 LRM series (Yang et al., 2025) against the specialized xLAM2 series (Prabhakar et al., 2025), including a "no-think" Qwen3 baseline to isolate reasoning effects. As shown in Figure 2, standard LRMs outperform both the "no-think" baseline and specialized models on parallel and irrelevance tasks. This indicates that explicit reasoning effectively captures the structural dependencies between the query $Q$ and tool set $T$. However, in complex multi-turn scenarios, LRMs lag behind specialized models. This significant performance gap motivates our investigation.

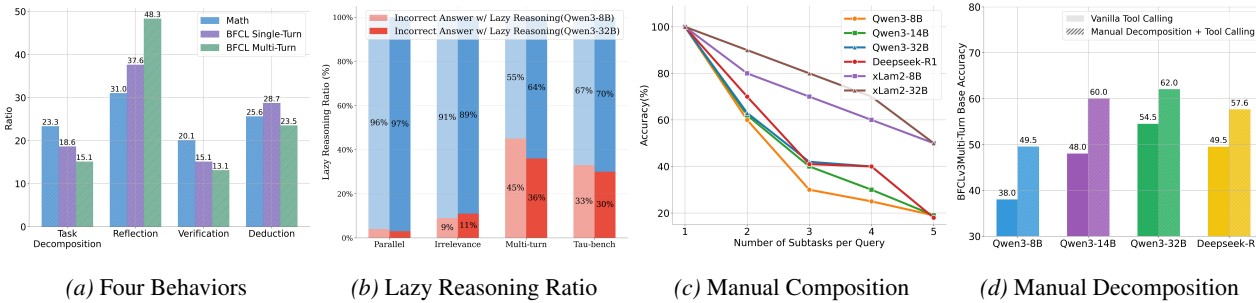

*(a)* Four Behaviors   *(b)* Lazy Reasoning Ratio   *(c)* Manual Composition   *(d)* Manual Decomposition

*Figure 3.* **(a)** Distribution of four behavioral categories across rollout samples. **(b)** Ratio of lazy reasoning in different tasks.**(c)** Accuracy degradation caused by subtask composition.**(d)** Accuracy improvement through manual task decomposition.

### 2.3. Lazy Reasoning in Tool Use

**Reasoning behavior variations in tool use.** To investigate the deficiency in multi-turn scenarios, we sample 20 rollouts per query on BFCL v3 using Qwen3-8B, employing MATH-500 (Hendrycks et al., 2021) as a reference baseline (Ning et al., 2025; Bogdan et al., 2025; Venhoff et al., 2025). Following Ning et al. (2025), we parse each reasoning process into thoughts and categorize them into four types: (i) **Task Decomposition** (breaking down problems), (ii) **Reflection** (revising errors), (iii) **Verification** (checking results), and (iv) **Deduction** (inferring from assumptions).

Figure 3a reveals a distinct contrast: compared to single-turn or math tasks, multi-turn reasoning exhibits minimal task decomposition yet excessive reflection.This implies a mode of ineffective computation: the model expends significant capacity on verbose generation while bypassing substantive structural reasoning. We term this phenomenon **Lazy Reasoning**. We quantify this behavior based on empirical thresholds for token generation and reflection frequency. As shown in Figure 3b, the prevalence of Lazy Reasoning correlates strongly with failures in multi-turn interactions. Further implementation details are provided in Appendix A.1.

**Addressing Lazy Reasoning.** Recent analysis (Patil et al.) attributes multi-turn failures to deficits in planning and execution tracking. This aligns with our observation. We hypothesize that Lazy Reasoning is a compensatory mechanism: models default to unproductive trial-and-error when they lack the capacity for structural decomposition. To verify this, we construct multi-subtask queries by composing single-subtask ones (Chen et al., 2024). Figure 3c shows that performance degrades as the number of subtasks increases. This observation implies that decomposition complexity is a primary inducer of Lazy Reasoning.

We therefore revisit task decomposition as a remedy. Classical paradigms, such as Decomposed Prompting (Khot et al., 2022b) and Least-to-Most Prompting (Zhou et al., 2022), operate via a "plan-and-execute" mechanism. We adopt this strategy to provide explicit structural guidance, com-

pelling the model to strictly follow step-by-step execution (§A.2.5). As demonstrated in Figure 3d, our hypothesis is strongly corroborated: imposing appropriate decomposition structure acts as a catalyst for reasoning, effectively counteracting Lazy Reasoning and unlocking significant gains in multi-turn LRM scenarios.

### 3. D-CORE

Building on the effectiveness of decomposition-based prompting, we propose **D-CORE**, a framework that enables LRMs to autonomously master complex tool use. Our method operates in two stages: (i) self-distillation to incentivize task decomposition and subtask execution, and (ii) diversity-aware GRPO to stabilize reinforcement learning and enhance reflective reasoning.

### 3.1. Incentivize Complex Task Decomposition Reasoning via Self-Distillation.

While task decomposition is known to improve tool use, injecting this capability typically relies on powerful "teacher" LRMs to synthesize reasoning trajectories (Huang et al., 2022; Schick et al., 2023a; Madaan et al., 2023). We challenge this reliance on external supervision. We observe that *current LRMs inherently possess the capacity to generate high-quality reasoning trajectories when provided with explicit structural guidance*. Based on this insight, our self-distillation framework elicits decomposition capabilities directly from the model itself, eliminating the need for a stronger teacher.

**Task Decomposition.** We sample seed datasets and prompt LRM $\mathcal{M}$ to decompose queries $Q$ into subtasks $\mathcal{S} \leftarrow$ Decompose$(\mathcal{C}, Q, Y^*, \mathcal{M})$, where $\mathcal{C} = \{P, T, C\}$ represents comprehensive contextual information: system policy $P$, tool set $T$, conversation history $C$. The reference trajectories $Y^*$ are provided along with few-shot examples to improve decomposition success rates, serving as the most critical factor for enhancing task decomposition performance. The complete prompt can be found in Appendix A.3.

**Reasoning Generation.** Given the decomposed subtasks

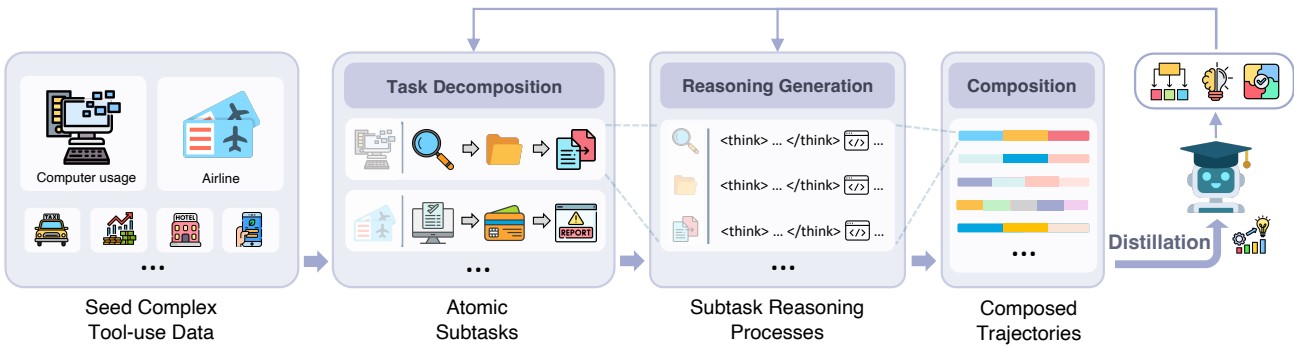

*Figure 4.* Overview of D-CORE self-distillation stage. Through self-distillation, a LRM with task decomposition and subtask execution capabilities is acquired.

$\mathcal{S} = \{s_1, s_2, \ldots, s_n\}$, we feed them into LRM $\mathcal{M}$ to generate reasoning processes and tool calls $(\mathcal{R}_i, \tau_i) \leftarrow \mathcal{M}(\mathcal{C}, s_i)$. The generation process varies by scenario: sequential subtasks are processed iteratively to maintain context dependency, while parallel subtasks are handled simultaneously. For tool-irrelevant queries where decomposition are not applicable, we prompt $\mathcal{M}$ to generate explainations for why task cannot be decomposed.

**Composition.** At composition stage, for sequential and parallel scenario, we compose subtasks $\mathcal{S}$, reasoning process $\mathcal{R}$, tool calls $\tau$ and tool response $o$ into complete reasoning trajectories $\hat{\mathcal{Y}} \leftarrow \text{Compose}(\{(s_i, \mathcal{R}_i, \tau_i, o_i)\}_{i=1}^{|\mathcal{S}|})$ while $\hat{\mathcal{Y}} \leftarrow \text{Compose}(\mathcal{R}, Y^*)$ for tool-irrelevant scenario. Reflection mechanisms are incorporated in Compose template for parallel and irrelevant scenarios.

**Distillation.** Based on the composed reasoning trajectories $\hat{\mathcal{Y}}$, we apply SFT on LRM $\mathcal{M}$. During the SFT, LRM acquire task decomposition and subtask execution capabilities through parameters $\theta$ by maximizing the probability $\pi_\theta(\hat{\mathcal{Y}}_t \mid (\mathcal{C}, Q), \hat{\mathcal{Y}}_{1:t-1})$, the optimization loss function can be expressed as follows:

$$\mathcal{L}_{\text{self-distillation}}(\theta) = -\mathbb{E}[\log \pi_\theta(\hat{\mathcal{Y}}_t \mid (\mathcal{C}, Q), \hat{\mathcal{Y}}_{1:t-1})]. \quad (1)$$

In our self-distillation framework, the LRM $\mathcal{M}$ generates subtasks and corresponding reasoning traces. This process synthesizes a dataset that encapsulates the problem-solving logic of $\mathcal{M}$. We provide the specific prompts and construction details in the Appendix.

### 3.2. Diversity-Aware GRPO

Although self-distillation improves task decomposition, it suppresses reflective reasoning and exploration. This reduction in diversity is evidenced by the near-zero reward standard deviation of the self-distilled model (Figure 5a). Such low variance hinders GRPO optimization, where the advantage $A_{i,t}$, which depends on the deviation from mean reward, becomes negligible when all rewards are nearly

---

**Algorithm 1** Decomposing Task and Composing Reasoning

**Input:** Context $\mathcal{C} = \{P, T, C\}$ where $P$ is system policy, $T$ is tool set, $C$ is conversation history, query $Q$, LRM $\mathcal{M}$, reference trajectories $Y^*$

**Output:** Composed reasoning trajectories $\hat{\mathcal{Y}}$

1: $\mathcal{S} \leftarrow \text{Decompose}(\mathcal{C}, Q, Y^*, \mathcal{M})$
2: **return** $\emptyset$ **if** $|\mathcal{S}| \neq Y^*$
3: **if** $\mathcal{S}$ is SEQUENTIAL **then**
4:     $\mathcal{I}_0 \leftarrow \text{InitInput}(\mathcal{C})$
5:     **for** $i = 1, \ldots, |\mathcal{S}|$ **do**
6:         $(\mathcal{R}_i, \tau_i) \leftarrow \mathcal{M}(\mathcal{I}_{i-1}, s_i)$
7:         $o_i \leftarrow \text{Execute}(\tau_i)$
8:         $\mathcal{I}_i \leftarrow \text{Update}(\mathcal{I}_{i-1}, \tau_i, o_i)$
9:     **end for**
10:     $\hat{\mathcal{Y}} \leftarrow \text{Compose}_{\text{seq}}(\{(s_i, \mathcal{R}_i, \tau_i, o_i)\}_{i=1}^{|\mathcal{S}|})$
11: **else if** $\mathcal{S}$ is PARALLEL **then**
12:     **for** $i = 1, \ldots, |\mathcal{S}|$ **do**
13:         $(\mathcal{R}_i, \tau_i) \leftarrow \mathcal{M}(\mathcal{C}, s_i)$
14:     **end for**
15:     $\hat{\mathcal{Y}} \leftarrow \text{Compose}_{\text{par}}(\{(s_i, \mathcal{R}_i, \tau_i)\}_{i=1}^{|\mathcal{S}|})$
16: **else**
17:     $\mathcal{R} \leftarrow \mathcal{M}(\mathcal{C}, Q)$
18:     $\hat{\mathcal{Y}} \leftarrow \text{Compose}_{\text{irr}}(\mathcal{R}, Y^*)$
19: **end if**
20: **return** $\hat{\mathcal{Y}}$ **if** $\text{Verify}(\hat{\mathcal{Y}}, Y^*)$ **else** $\emptyset$

---

identical:

$$A_{i,t} = \frac{R_i - \text{mean}(\{R_i\}_{i=1}^G)}{\text{std}(\{R_i\}_{i=1}^G)}, \quad (2)$$

where $G$ is the rollouts number. Omitting the KL penalty, the GRPO objective is formulated as:

$$\mathcal{J}_{\text{GRPO}}(\theta) = \mathbb{E}[\min(r_{i,t} A_{i,t}, \text{clip}(r_{i,t}, 1 - \epsilon, 1 + \epsilon) A_{i,t})]. \quad (3)$$

For clarity, we omit the clipping term in subsequent derivations. The resulting policy gradient of GRPO can be ex-

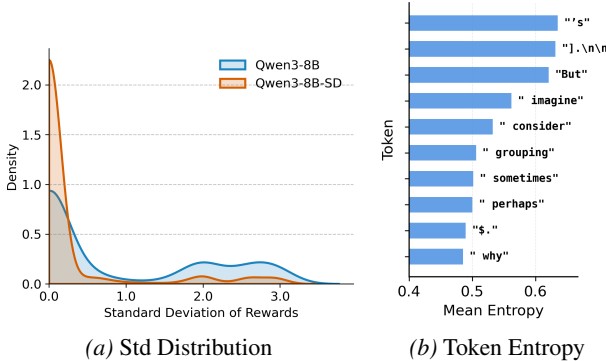

*(a) Std Distribution*  *(b) Token Entropy*

*Figure 5.* **(a)** Effect of self-distillation on reward variance. SD: self-distillation. **(b)** Top 10 high-entropy tokens for Qwen3-8B-SD.

pressed as:

$$\nabla_\theta \mathcal{J}_{\text{GRPO}} = \frac{1}{G}\left(\sum_{(i,t)} \frac{1}{|y_i|} r_{i,t}(\theta) A_{i,t} \nabla_\theta \log \pi_\theta(y_{i,t}|x_i, y_{i,<t})\right), \tag{4}$$

where

$$r_{i,t}(\theta) = \frac{\pi_\theta(y_{i,t} \mid x_i, y_{i,<t})}{\pi_{\text{old}}(y_{i,t} \mid x_i, y_{i,<t})}, \tag{5}$$

We observe a critical degeneracy: as the generation variance diminishes, the advantage $A_{i,t}$ vanishes, causing the gradient signal to collapse. This necessitates an alternative formulation. Motivated by recent evidence that high-entropy tokens correlate with reflective reasoning (Wang et al., 2025; Cheng et al., 2025)—a behavior we corroborate in Figure 5b—we investigate the following question: *Can an entropy-aware advantage formulation mitigate gradient collapse and sustain reasoning diversity?*

Guided by this observation, we propose **Diversity-Aware GRPO**. By reshaping the advantage with an entropy-based term, our method mitigates gradient collapse while explicitly incentivizing complex reasoning. The formulation is defined as:

$$\hat{A}_{i,t} = \begin{cases} \psi(\mathcal{H}_{i,t}) & \text{if } A_{i,t} < \zeta, \\ A_{i,t} & \text{otherwise,} \end{cases} \tag{6}$$

where $\zeta$ is a small constant for numerical stability (e.g., $\zeta = 10^{-8}$) and $\psi(\mathcal{H}_{i,t})$ is an entropy-based advantage term:

$$\psi(\mathcal{H}_{i,t}) = \min(\alpha \cdot \mathcal{H}_{i,t}^{\text{detach}}, \delta), \tag{7}$$

in which larger $\alpha$ promotes high-entropy token generation, while $\delta$ bounds the entropy advantage. The entropy term $\mathcal{H}_{i,t}^{\text{detach}}$ is detached during backpropogation, serving as a fixed offset that adjust update magnitude without affecting gradients. It is computed over the token vocabulary $\mathcal{V}$ as:

$$\mathcal{H}_{i,t}^{\text{detach}} = -\sum_{v \in \mathcal{V}} \pi_\theta(v|x_i, y_{i,<t}) \log \pi_\theta(v|x_i, y_{i,<t}). \tag{8}$$

The key idea is to scale each token's gradient update in proportion to its detached entropy $\mathcal{H}_{i,t}^{\text{detach}}$ when std approaches zero. The objective function of DA-GRPO is formulated as:

$$\mathcal{J}_{\text{DA-GRPO}}(\theta) = \mathbb{E}[\min(r_{i,t}(\theta)\hat{A}_{i,t}, \text{clip}(r_{i,t}(\theta), 1-\epsilon, 1+\epsilon)\hat{A}_{i,t}) \\ - \lambda \mathbb{D}_{\text{KL}}[\pi_\theta || \pi_{\text{ref}}]]. \tag{9}$$

Omitting the KL divergence term, the policy gradient of DA-GRPO is formulated as:

$$\nabla_\theta \mathcal{J}_{\text{DA-GRPO}} = \mathbb{E}[r_{i,t}(\theta)\hat{A}_{i,t}\nabla_\theta \log \pi_\theta(y_{i,t}|x_i, y_{i,<t})]. \tag{10}$$

We partition the sample indices into two disjoint sets based on the advantage values:

$$\mathcal{T}_{\neq 0} = \{(i,t) : \hat{A}_{i,t} \neq 0\}, \tag{11}$$
$$\mathcal{T}_{=0} = \{(i,t) : \hat{A}_{i,t} = 0\}. \tag{12}$$

Accordingly, the policy gradient can be decomposed into two distinct components:

$$\nabla_\theta \mathcal{J}_{\text{DA-GRPO}} = \nabla_\theta \mathcal{J}_{\text{GRPO}} \\ + \underbrace{\frac{1}{G} \sum_{(i,t) \in \mathcal{T}_{=0}} \frac{1}{|y_i|} r_{i,t}(\theta)\psi(\mathcal{H}_{i,t})\nabla_\theta \log \pi_\theta(y_{i,t}|x_i, y_{i,<t})}_{\text{Entropy Advantage Term}}. \tag{13}$$

**Theorem 3.1** (Prevention of Learning Stagnation). *Let* $\mathcal{T}_{=0} \neq \emptyset$ *be the set of positions where* $A_{i,t} = 0$. *If there exists* $(i,t) \in \mathcal{T}_{=0}$ *such that:*

*1.* $r_{i,t}(\theta) \neq 0$

*2.* $\psi(\mathcal{H}_{i,t}) > 0$ *(i.e.,* $\pi_\theta(\cdot|q, o_{i,<t})$ *is not degenerate)*

*Then* $\|\nabla_\theta \mathcal{J}_{\text{DA-GRPO}}\| > 0$, *ensuring continued learning.*

**Theorem 3.2** (Entropy Reduction Property of DA-GRPO). *When* $A_{i,t} = 0$ *for some token position* $(i,t)$, *DA-GRPO encourages the generation of high-entropy tokens by reducing their entropy. Specifically, for* $(i,t) \in \mathcal{T}_{=0}$, *the gradient contribution is:*

$$\nabla_\theta \mathcal{J}_{i,t} = \frac{1}{G} \sum_{(i,t)} \frac{1}{|y_i|} r_{i,t}(\theta)\psi(\mathcal{H}_{i,t})\nabla_\theta \log \pi_\theta(y_{i,t}|x_i, y_{i,<t}) \tag{14}$$

*, where* $\mathcal{H}_{i,t} = -\log \pi_{\theta_{old}}(y_{i,t}|x_i, y_{i,<t})$ *is the detached cross-entropy. Since* $r_{i,t}(\theta) > 0$ *and tokens with higher* $\mathcal{H}_{i,t}$ *(lower probability under* $\pi_{\theta_{old}}$*) receive stronger positive gradients, DA-GRPO preferentially increases the probability of high-entropy tokens, thereby reducing their entropy and making them more likely to be generated.*

The reward employed in DA-GRPO aligns with that of ToolRL (Qian et al., 2025)(Appendix A.2):

$$R_i = R_{\text{format}} + R_{\text{struct}} + R_{\text{key}} + R_{\text{value}}. \tag{15}$$

*Table 1.* Comparison of the accuracy on BFCLv3 and $\tau$-Bench. The best results are **bolded**.

| Model | BFCLv3 | | | | | | $\tau$-Bench | | |
|---|---|---|---|---|---|---|---|---|---|
| | Average | Live | Non-Live | Relevance | Irrelevance | Multi-Turn | Average | Retail | Airline |
| *Proprietary Model Results* | | | | | | | | | |
| Claude-3.7-Sonnet | 58.6 | 78.4 | 41.3 | 81.4 | 72.2 | 48.4 | 59.8 | **78.3** | 41.2 |
| DeepSeek-R1 | 63.8 | 77.3 | 75.5 | 73.3 | 77.8 | 38.9 | 47.8 | 55.6 | 40.0 |
| o1 | 67.8 | 80.6 | 85.7 | 87.8 | 72.2 | 36.1 | **63.9** | 73.5 | **54.2** |
| GPT-4o | 71.7 | 78.8 | 86.8 | 81.3 | 83.3 | 50.0 | 52.9 | 62.8 | 43.0 |
| *Open-Source Model Results* | | | | | | | | | |
| Qwen3-8B | 66.3 | 78.5 | 88.8 | 79.1 | 77.8 | 33.0 | 29.0 | 34.7 | 23.2 |
| Qwen3-14B | 65.9 | 80.0 | 88.2 | 81.0 | 72.2 | 36.6 | 33.6 | 41.6 | 25.6 |
| Qwen3-32B | 69.2 | 77.3 | 88.9 | 75.8 | 72.2 | 43.1 | 36.6 | 39.6 | 33.6 |
| xLAM2-8B | 72.0 | 66.7 | 84.4 | 64.3 | 77.8 | 69.1 | 42.4 | 50.7 | 34.0 |
| xLAM2-32B | 76.4 | 74.2 | 89.3 | 76.7 | **88.9** | 67.1 | 45.0 | 52.5 | 37.6 |
| xLAM2-70B | 78.4 | 72.9 | 88.4 | 78.9 | 66.7 | **75.0** | 48.3 | 57.7 | 38.8 |
| *Custom-Trained Model Results* | | | | | | | | | |
| ToolRL-Qwen3-8B | 65.9 | 82.4 | 88.9 | 85.9 | 77.8 | 26.8 | 31.5 | 40.6 | 22.4 |
| ToolRL-Qwen3-14B | 68.5 | 81.5 | **89.4** | 81.9 | 72.2 | 35.8 | 37.8 | 49.5 | 26.0 |
| D-CORE-8B(ours) | 77.7 | 82.2 | 86.4 | 88.2 | 77.8 | 63.8 | 47.6 | 50.7 | 44.4 |
| *Δ from Qwen3-8B* | +11.4 | +3.7 | -2.4 | +9.1 | 0.0 | +30.8 | +18.6 | +16.0 | +21.8 |
| D-CORE-14B(ours) | **79.3** | **82.9** | 87.2 | **89.2** | 72.2 | 67.4 | 51.3 | 56.5 | 46.0 |
| *Δ from Qwen3-14B* | +13.4 | +2.9 | -1.0 | +8.4 | 0.0 | +30.8 | +17.7 | +14.9 | +20.4 |

## 4. Experiments

### 4.1. Experimental Setups

We collect data from two sources: subsets of open-source datasets (Liu et al., 2024a;b; Prabhakar et al., 2025) and trajectories of our custom-built tool use agent, ensuring coverage across diverse domains and difficulty levels. From these sources, we generate 40,000 self-distillation instances and sample 5,000 instances for RL. To facilitate reproducibility, we release the full dataset and the data generation pipeline. We use Qwen3-8B/14B (Yang et al., 2025) as the backbone LRM, with packing technique (Xu et al., 2024) for self-distillation and verl framework (Sheng et al., 2024) for RL training. We evaluate on two realistic agent benchmarks: BFCLv3 (Patil et al.) and $\tau$-bench (Yao et al., 2024). We compare against open-source models (Guo et al., 2025; Liu et al., 2024a; Prabhakar et al., 2025), as well as closed-source LRMs and LLMs (OpenAI, 2024b; Anthropic, 2025; DeepMind, 2024; OpenAI, 2024a). We implement the data generation pipeline (Algorithm 1) on a single NVIDIA A100 GPU. Generating 40k samples takes 25 hours (8B) and 30 hours (14B). The D-CORE training requires 22 hours (8B) and 38 hours (14B). Both self-distillation and DA-GRPO are trained for 3 epochs on 8 x A100. Details are provided in Appendix A.6.

### 4.2. Main Results

$\tau$-**bench.** As shown in Table 1, D-CORE achieves substantial gains on $\tau$-bench, improving accuracy by 18.6% for Qwen3-8B and 17.7% for Qwen3-14B, validating the effectiveness and scalability of our approach. Notably, D-CORE-14B excels in the $\tau$-airline task with the highest

accuracy of 46.0%, where the LRM handles complex refund evaluation and compensation decisions, requiring 4-5 subtasks per query when user intentions are unclear.

**BFCLv3.** Table 1 reveals that D-CORE achieves substantial accuracy gains of 11.4% and 13.4% for Qwen3-8B and Qwen3-14B respectively, with remarkable improvements of 30.8% on challenging multi-turn tasks. In contrast, the baseling ToolRL method using GRPO alone fails to deliver meaningful improvements in multi-turn scenarios, highlighting the effectiveness of our approach. D-CORE-8B establishing new state-of-the-art among 8B models at 77.7% overall accuracy, significantly outperforming xLAM2-8B. D-CORE-14B achieves 79.3% overall accuracy with 5× fewer parameters than the previous state-of-the-art 70B LLM, validating LRM's test-time scaling effectiveness.

### 4.3. Ablation Study and Analysis

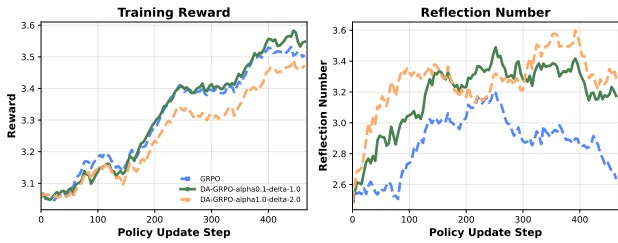

*Figure 6.* Training dynamics comparison across GRPO and DA-GRPO on D-CORE-8B.

**Training dynamics.** Figure 6 shows the training dynamics of D-CORE-8B. With $\alpha$=0.1, DA-GRPO achieves the high-

*Table 2.* Accuracy on out-of-distribution tasks. The best results are **bolded**.

| Model | $\tau^2$-Bench | | | | ACEBench-en | | | | BFCLv4-Agentic | |
|---|---|---|---|---|---|---|---|---|---|---|
| | Retail | Airline | Telecom | Overall | Normal | Special | Agent | Overall | Web-base | Memory |
| Qwen3-8B | 41.5 | 31.3 | 26.3 | 33.0 | 71.4 | 75.3 | 29.1 | 65.9 | 16.0 | 18.9 |
| Qwen3-14B | 46.5 | 30.0 | 31.7 | 36.1 | 66.9 | 84.0 | 44.2 | 68.0 | 34.0 | 25.2 |
| Qwen3-32B | 49.1 | 36.0 | 28.4 | 37.8 | 75.9 | 77.3 | 49.2 | 72.2 | 32.0 | 25.0 |
| xLAM2-8B | 51.3 | 35.4 | 22.4 | 36.4 | 58.8 | 0.0 | 5.0 | 34.8 | 8.0 | 15.7 |
| xLAM2-32B | 53.1 | 41.6 | 26.1 | 40.3 | 69.2 | 24.7 | 13.4 | 52.5 | 31.0 | 16.8 |
| xLAM2-70B | **54.9** | **46.0** | 29.3 | 43.4 | 57.1 | 5.3 | 38.4 | 36.5 | 13.0 | 17.6 |
| D-CORE-8B (ours) | 49.8 | 38.4 | 30.5 | 39.6 | **77.9** | 78.7 | **59.2** | 75.2 | 36.0 | 20.4 |
| D-CORE-14B (ours) | 53.5 | 44.1 | **34.9** | **44.2** | 77.1 | **88.7** | 55.8 | **76.9** | **39.0** | **26.5** |

est reward while consistently increasing reflection tokens. At $\alpha$=1.0, reflection tokens increase progressively but rewards remain lower than GRPO. This revealing a trade-off: entropy-based advantages encourage reflection, but excessive $\alpha$ causes exploration collapse.

**Generalizability of D-CORE.** We validate out-of-distribution generalization on three diverse benchmarks: ACEBench (Chen et al., 2025a), $\tau^2$-Bench (Barres et al., 2025) and BFCLv4-agentic (Patil et al.). $\tau^2$-Bench challenges agents in dual-control environments with shared world states. ACEBench, characterized by complex system and user prompts, serves as a stress test for generalization. It reveals a critical limitation in standard fine-tuning: SFT models often degrade in generalization, underperforming open-source general-purpose models. Our results demonstrate that D-CORE effectively mitigates this issue, surpassing the Qwen3 baseline. We further extend the evaluation to BFCLv4-agentic, which introduces scenarios for Web-search and Memory. As shown in Table 2, D-CORE remains highly competitive on these unseen, complex tasks. This consistency confirms that our performance gains stem from intrinsic improvements in task decomposition and reasoning, rather than overfitting to specific training distributions.

| Model | BFCLv3 MT(%) | $\tau$-bench Overall(%) |
|---|---|---|
| Qwen3-8B | 33.0 | 29.0 |
| +GRPO | 26.8(-6.2) | 31.5(+2.5) |
| +SD$_{n=10,000}$ | 51.5 | 22.1 |
| +SD$_{n=20,000}$ | 53.1 | 37.8 |
| +SD$_{n=40,000}$ | 57.5 | 36.6 |
| +GRPO | 67.4(+9.9) | 45.2(+8.6) |

*Table 4.* Effect of self-distillation on multi-turn tool use capability. SD: self-distillation; MT: Multi-Turn; n: number of training samples.

**Effectiveness of Self-Distillation.** As shown in Table 4, self-distillation demonstrates improved effectiveness with increasing sample size and mitigates RL optimization challenges. Notably, when GRPO is applied to Qwen3-8B, improvements are marginal or even negative. In contrast, a self-distilled model achieves substantial gains of 9.9% and 8.6% on the respective benchmarks under identical settings,

highlighting the efficacy of our approach.

**Quality of D-CORE Reasoning Trajectories.** To evaluate the effectiveness of D-CORE generated trajectories (Algorithm 1), we conduct SFT on Llama3.1-8B and Qwen2.5-14B using trajectories produced by Qwen3-8B. As shown in Table 5, models trained with our trajectories significantly outperform Deepseek-R1 SFT on complex tool-use scenarios.

| Model | BFCLv3 overall(%) | $\tau$-bench overall(%) |
|---|---|---|
| DS-R1-Llama3.1-8B | 27.8 | 19.0 |
| D-CORE-Llama3.1-8B | 63.7 | 36.0 |
| DS-R1-Qwen2.5-14B | 47.9 | 18.2 |
| D-CORE-Qwen2.5-14B | 70.5 | 41.5 |

*Table 5.* Performance comparison of trajectory-distilled reasoning models: Deekseek-R1 vs. D-CORE (Qwen3-8B).

**Effectiveness of Task Decomposition.** We evaluate the impact of task decomposition strategies using the Qwen3-8B model (Table 6). While relying on ground-truth reference trajectories or few-shot examples yields optimal decomposition, such supervision is often inaccessible in complex, real-world scenarios. To address this, we explore a more scalable alternative: leveraging pseudo-labels generated by a stronger model (Qwen3-Max). Our results demonstrate that this approach maintains high effectiveness, bridging the gap between ideal supervision and practical applicability.

| Ref Traj | Few Shot | Traj Type | Success Rate(%) | $\tau$-bench overall(%) | BFCLv3 overall(%) |
|---|---|---|---|---|---|
| ✗ | ✗ | ✗ | 73.8 | 20.3 | 68.4 |
| ✓ | ✗ | GT | 89.1 | 33.8 | 68.3 |
| ✓ | ✓ | GT | 93.2 | 37.1 | 68.4 |
| ✓ | ✓ | PL | 92.8 | 29.6 | 68.5 |

*Table 6.* Task decomposition success rates based on Qwen3-8B. GT: ground truth. PL: pseudo-labels.

*Table 3.* Accuracy of BFCLv3 and $\tau$-Bench, where SD represents self-distillation. Avg: The average score of BFCLv3 and $\tau$-Bench, indicating the average expected gain. The best results are **bolded**.

| Model | Avg | BFCLv3 | | | | | $\tau$-Bench | |
|---|---|---|---|---|---|---|---|---|
| | Accuracy | Live | Non-Live | Relevance | Irrelevance | Multi-Turn | Retail | Airline |
| Qwen3-8B | 38.1 | 78.5 | **88.8** | 77.8 | 79.1 | 33.0 | 34.7 | 23.2 |
| +SD | 46.2 | **82.5** | 86.5 | 66.7 | **89.7** | 57.5 | 42.0 | 31.2 |
| +GRPO | 55.6 | 75.6 | 85.4 | 72.2 | 84.5 | 67.4 | 49.2 | 41.2 |
| *Δ from SD* | +9.4 | -6.9 | -1.1 | +5.5 | -5.2 | +9.9 | +7.2 | +10.0 |
| +DA-GRPO$_{\alpha=0.01,\delta=0.5}$ | 56.2 | 77.0 | 86.7 | 77.8 | 83.7 | **67.9** | 52.5 | 38.8 |
| +DA-GRPO$_{\alpha=0.1,\delta=0.5}$ | **57.6** | 82.3 | 86.4 | 77.8 | 89.4 | 63.8 | 50.7 | **44.4** |
| *Δ from SD* | +11.4 | -0.2 | -0.1 | +11.1 | -0.3 | +6.3 | +8.7 | +13.2 |
| +DA-GRPO$_{\alpha=0.4,\delta=0.5}$ | 55.5 | 78.8 | 86.1 | 72.2 | 85.5 | 59.9 | **54.6** | 36.8 |
| +DA-GRPO$_{\alpha=1.0,\delta=1.0}$ | 54.0 | 76.4 | 84.8 | **83.3** | 84.4 | 62.1 | 47.9 | 39.6 |

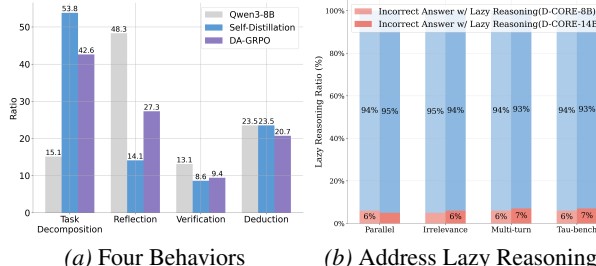

*(a)* Four Behaviors     *(b)* Address Lazy Reasoning

*Figure 7.* **(a)** Behavioral distribution changes after self-distillation and DA-GRPO. **(b)** Lazy Reasoning ratios across tasks for the D-CORE models.

**Mitigating the Lazy Reasoning.** Figure 7b demonstrates that the D-CORE model reduces the proportion of Lazy Reasoning in incorrect answers for BFCLv3 Multi-Turn and $\tau$-bench to levels comparable with Parallel and Irrelevance categories. As a direct control, the 8B model shows a reduction in errors caused by Lazy Reasoning from 45% to 6% in Multi-Turn tasks, demonstrating the effectiveness of our approach in mitigating the Lazy Reasoning phenomenon.

**Effectiveness of DA-GRPO.** DA-GRPO addresses a central failure mode of post-distillation RL: collapsed reward variance. As shown in Table 7, standard exploration strategies are insufficient. Increasing the sampling temperature often breaks the rigid tool-use syntax before producing useful reasoning diversity. Mixed-policy GRPO rollouts (Yan et al., 2026) creates simple advantage gaps, but mainly reinforces existing "lazy" behaviors rather than inducing new decomposition strategies, as illustrated in Fig. 3a. DA-GRPO takes a more structured approach. It decouples semantic exploration from syntactic correctness by injecting entropy only into uncertain reasoning tokens. This targeted exploration restores reflective reasoning, which is largely suppressed by self-distillation, as shown in Fig. 7a, while preserving accurate tool calls. Consequently, DA-GRPO consistently outperforms strong RL baselines, including RL-ZVP (Le et al., 2025), DAPO (Yu et al., 2026), and Seed-GRPO (Chen

et al., 2025b), in Table 8. Although Seed-GRPO benefits from semantic entropy, it does not fully resolve advantage collapse. DA-GRPO does, leading to simultaneous improvements in tool correctness and deep reflection, as reported in Table 3. Finally, this exploration must be properly regularized: $\alpha = 0.1$ gives the best exploration–exploitation trade-off, whereas excessive entropy weakens the reward signal; see Appendix A.5.

| Method | $\tau$-bench overall(%) | BFCLv3 overall(%) | Avg (%) |
|---|---|---|---|
| GRPO($\tau$=1.0) | 45.2 | 78.4 | 61.8 |
| GRPO($\tau$=2.0) | 45.5 | 77.5 | 61.5 |
| Mixed-policy-GRPO | 45.0 | 77.1 | 61.1 |
| DA-GRPO | 47.6 | 77.7 | 62.7 |

*Table 7.* **Exploration strategies for post-distillation RL.** Naive remedies-high temperature $\tau$ or mixing undistilled rollouts-fail to surpass the GRPO baseline.

| Method | $\tau$-bench overall(%) | BFCLv3 overall(%) | Avg (%) |
|---|---|---|---|
| xLAM2 | 42.4 | 72.0 | 57.2 |
| GRPO | 45.2 | 78.4 | 61.8 |
| GRPO + DS | 43.6 | 77.8 | 60.7 |
| GRPO + RL-ZVP | 44.0 | 78.5 | 61.3 |
| Seed-GRPO | 45.2 | 78.5 | 61.9 |
| DA-GRPO | 47.6 | 77.7 | 62.7 |

*Table 8.* **Comparisons with recent RL enhancements on tool use tasks.** DA-GRPO consistently outperforms standard GRPO and its recent variants.

**Sensitivity to Lazy Reasoning thresholds.** Lazy Reasoning is governed by two hyperparameters: the reflection number and response length. We evaluate 6 combinations across a spectrum of values (Length $\in \{200, 300, 400\}$, Reflection number $\in \{3, 4, 5, 6\}$ ) and report results in Table 9. Three

observations emerge: *(i) Ranking is invariant.*The category ordering Multi-Turn $\gg$ Irrelevance $>$ Parallel holds across all settings (Kendall's $W = 1.0$). *(ii) Separation is large.* Multi-Turn diverges from other categories by Cohen's $d$=6.7, well beyond standard significance thresholds. *(iii) Concentration is stable.* The fraction of Lazy Reasoning within Multi-Turn failures remains at $76.3\%\pm0.9\%(CV = 1.2\%)$. Together, these results indicate that the phenomenon is intrinsic to the model's behavior rather than an artifact of threshold choice.

| Reflection Number | Length | Multi Turn(%) | Parallel (%) | Irrelevance (%) |
|---|---|---|---|---|
| 3 | 200 | 45.1 | 3.7 | 9.1 |
| 3 | 300 | 35.6 | 3.7 | 7.8 |
| 3 | 400 | 35.6 | 3.7 | 7.8 |
| 4 | 200 | 39.7 | 3.7 | 8.3 |
| 5 | 200 | 33.5 | 3.7 | 7.5 |
| 6 | 200 | 30.8 | 2.3 | 7.3 |

*Table 9.* **Sensitivity to Lazy Reasoning thresholds.** Lazy Reasoning ratios (%) across reflection numbers and response lengths. The ranking Multi-Turn $\gg$ Irrelevance $>$ Parallel is *invariant* to threshold choices.

**Training Cost.** We compare the end-to-end training time of D-CORE and ToolRL in Table 10. Self-distillation comprises two stages: data generation and SFT. Data generation is performed with vLLM on a single A100 GPU, taking 25 and 30 hours to produce 40k samples for Qwen3-8B and Qwen3-14B, respectively. SFT runs for 3 epochs on 8 A100 GPUs, taking 11 hours for Qwen3-8B and 21 hours for Qwen3-14B. Despite this overhead, D-CORE yields substantially better accuracy on both benchmarks, offering a favorable cost-performanc trade-off (details in Appendix A.6).

| Method | Stages | Computational Overhead(hours) | BFCLv3 (%) | $\tau$-bench (%) |
|---|---|---|---|---|
| ToolRL-Qwen3-8B | RL | 11 | 65.9 | 31.5 |
| D-CORE-8B | SD + RL | 47 | 77.7 | 47.6 |
| ToolRL-Qwen3-14B | RL | 17 | 68.5 | 37.8 |
| D-CORE-14B | SD + RL | 68 | 79.3 | 51.3 |

*Table 10.* **Training cost comparisons.**While D-CORE incurs higher computational overhead due to the self-distillation (SD) stage, it achieves substantial performance gains over the RL-only baseline (ToolRL), offering a favorable cost-performance trade-off.

## 5. Related Work

**Tool Use.** Recent efforts have focused on creating and curating datasets to enhance LLMs tool use competencies (Patil et al., 2023; Liu et al., 2024a;b; Chen et al., 2024; Schick et al., 2023b). APIGen-MT (Prabhakar et al., 2025)

and Magnet (Yin et al., 2025) all aim to enhance LLMs' multi-turn tool use capabilities by constructing datasets that closely resemble real-world scenarios. To tackle the poor generalizability of models trained using SFT, ToolRL (Qian et al., 2025) and Nemotron-N1 (Zhang et al., 2025) applied long CoT and RL training methodologies to tool use tasks and conducted evaluations on BFCLv3 (Patil et al.) single-turn tasks. However, the analysis and improvement of long CoT and RL's effectiveness in enhancing tool use within complex scenarios (Yao et al., 2024) remains a challenging problem.

**Large Reasoning Models.** Large Reasoning Models represent a significant advancement in the evolution of language models (OpenAI, 2024b; 2025; Anthropic, 2025; Yang et al., 2025). DeepSeek-R1 (Guo et al., 2025) demonstrates that the GRPO (Shao et al., 2024) optimization algorithm combined with outcome reward mechanisms can enhance models' reasoning capabilities. A series of works have analyzed the impact of the reasoning process on outcomes from various perspectives (Ning et al., 2025; Hu et al., 2025; Gandhi et al., 2025; Shojaee et al., 2025). Additionally, some research on task-specific reasoning approach (Khot et al., 2022a; Zhou et al., 2022; Yao et al., 2023a;b; Besta et al., 2024) and Tool-Integrated Reasoning (Li et al., 2025a; Dong et al., 2025; Li et al., 2025b)has regained attention for improving current LRM training. We systematically transfer reasoning advances to complex tool use, yielding strong empirical gains.

## 6. Conclusion

In this work, we systematically study LRMs on tool use tasks. We identify that "Lazy Reasoning" in complex scenarios arises from insufficient task decomposition capabilities. We propose D-CORE to address this issue via self-distillation and DA-GRPO, achieving state-of-the-art results on BFCL-v3 and $\tau$-Bench.

## 7. Limitations and Future Work

While effective, D-CORE presents three natural avenues for future work. First, our SD+RL pipeline incurs a $4\times$ computational overhead, demanding more sample-efficient RL formulations. Second, validating whether this decoupling mechanism scales to larger regimes (e.g., 70B+) remains open, as current evaluations focus on $<=$14B models. Finally, we aim to generalize this paradigm beyond tool use to broader reasoning and multimodal contexts.

## Impact Statement

This paper aims to contribute to the advancement of Machine Learning through the exploration of Large Reasoning Models' tool use techniques. These techniques have the po-

tential to enhance model efficiency and scalability, leading to broader applicability across various domains. While there are numerous possible societal implications of our work, we do not identify any that require specific emphasis at this time.

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

# A. Appendix

## A.1. Details of Lazy Reasoning

The following cases clearly demonstrate the Lazy Reasoning phenomenon exhibited by Qwen3-8B when completing tasks on BFCLv3 and $\tau$-Bench. This manifests as the models' lack of detailed planning for the subtasks required to complete the main task, as well as the absence of specific reasoning for each subtask execution. Instead, the models display an inefficient, random pattern of trial-negation-retry thinking. Such reasoning processes occur in both correct and incorrect responses, indicating a cognitive pattern that models adopt when solving tool-use tasks, rather than an accidentally triggered behavior.

### A.1.1. CASE STUDY ON BFCLv3-MULTI-TURN 196

---

**Qwen3-8B**

{**"role"**: "user", **"content"**: "Gearing up for some exciting travel adventures! I wish to put a firm budget in place, setting a cap equivalent to 50,000 RMB for my upcoming European excursions. I'd appreciate if you could take charge of the conversion and establish a budget framework for me using the access token 'abc123xyz'."}

{**"role"**: "assistant", **"content"**: "`<think>` Okay, let's see. The user wants to set a budget limit of 50,000 RMB for their European travels and needs the conversion handled using the access token 'abc123xyz'.
First, I need to check which functions are available. The relevant ones here are probably related to the travel system since they mentioned budgeting and converting currency. Looking at the functions, there's ' set_budget_limit ' which requires an access token and a budget limit in USD. But the user provided the budget in RMB. So I need to convert 50,000 RMB to USD.
Wait, there's also 'compute_exchange_rate' which can convert between currencies. The parameters for that function include base_currency, target_currency , and value. The user's budget is in RMB ( base_currency : RMB), and they want it in USD ( target_currency : USD). The value is 50,000. So I should first call compute_exchange_rate with these parameters to get the USD equivalent. Once I have the converted USD amount, I can then use ' set_budget_limit ' with the access token 'abc123xyz' and the converted USD value to set the budget. But since the user mentioned put a firm budget in place, maybe they just want the limit set in RMB? Wait, the function ' set_budget_limit ' specifies the budget limit in USD. So the user's 50,000 RMB needs to be converted to USD first.
So the steps are:
1. Use compute_exchange_rate to convert 50,000 RMB to USD.
2. Use the resulting USD value with set_budget_limit .
But the user might not have mentioned the exchange rate, so I need to make sure to call compute_exchange_rate first. Let me check the parameters again. The compute_exchange_rate requires base_currency, target_currency , and value. The user's base is RMB, target is USD, value is 50000. Then, the response will give the exchanged_value in USD. Then, use that value in set_budget_limit . So the function calls would be: First, compute_exchange_rate( base_currency ='RMB', target_currency ='USD', value=50000). Then, set_budget_limit ( access_token ='abc123xyz', budget_limit =exchanged_value). But since the user's message doesn't mention the exchange rate, I need to perform the conversion first. Therefore, the correct sequence is to first call compute_exchange_rate, then use the result in set_budget_limit . However, the user might not have provided the exchange rate, so the assistant needs to handle that step.
`</think>`
[compute_exchange_rate( base_currency ='RMB', target_currency ='USD', value=50000),
 set_budget_limit ( access_token ='abc123xyz', budget_limit =exchanged_value)]}

---

## D-CORE-8B

{**"role"**: "user", **"content"**: "Gearing up for some exciting travel adventures! I wish to put a firm budget in place, setting a cap equivalent to 50,000 RMB for my upcoming European excursions. I'd appreciate if you could take charge of the conversion and establish a budget framework for me using the access token 'abc123xyz'."}

{**"role"**: "assistant", **"content"**:" `<think>` To complete this task, I need to break it down into the following subtasks:
1. Use the 'compute_exchange_rate' tool to convert 50,000 RMB to USD for budget setting, specifying 'RMB' as base currency and 'USD' as target currency.
2. Use the ' set_budget_limit ' tool to establish the converted USD amount as the budget framework using the provided access token 'abc123xyz'.
Now I should analyze the execution process of subtask 1. Okay, let's see. The user wants to convert 50,000 RMB to USD using the compute_exchange_rate tool. First, I need to check the parameters required for that function. The tool needs base_currency, target_currency , and value. The user specified RMB as the base and USD as the target, with a value of 50,000. So I should call compute_exchange_rate with those parameters. Let me make sure the parameters are correctly formatted. Base_currency is 'RMB', target_currency is 'USD', and value is 50000. That should do it. No other functions are needed here since the question is straightforward. ¡/think¿
[compute_exchange_rate( base_currency=\"RMB\", target_currency=\"USD\", value=50000)]

{**"role"**: "tool", **"content"** : "exchanged_value": 7142.86}

{**"role"**: "assistant", **"content"**:" `<think>` Now I should analyze the execution process of subtask 2:Use the ' set_budget_limit ' tool to establish the converted USD amount as the budget limit using access token 'abc123xyz'.
Okay, the user wants to set a budget limit using the converted USD amount from the previous exchange rate calculation. The exchange rate tool returned 7142.86 USD for 50,000 RMB. Now, they need to use the ' set_budget_limit ' function with the access token 'abc123xyz' and the budget limit as 7142.86.
First, I check the parameters required for ' set_budget_limit '. The required parameters are ' access_token ' and ' budget_limit '. The user provided the access token 'abc123xyz' and the converted value 7142.86.
I need to make sure the budget limit is a float, which it is. So the function call should be set_budget_limit ( access_token='abc123xyz', budget_limit =7142.86).
No other parameters are needed, and the function should execute correctly. The response will confirm the budget limit set in USD. `</think>`
[ set_budget_limit ( access_token=\"abc123xyz\", budget_limit =7142.86)]}

A.1.2. CASE STUDY ON τ-BENCH AIRLINE 6

---

**Qwen3-8B**

{**"role"**: "user", **"content"**: "My user ID is aarav_garcia_1177 . I don't have the reservation ID right now, but I want to change my flight to the cheapest economy option for the day after my original booking. I'm okay with the original payment being refunded as credit."},

{**"role"**: "assistant", **"content"**: "[ get_user_details ]"},

{**"role"**: "tool", **"content"**: " user_details "},

{**"role"**: "assistant", **"content"**: "`<think>` Okay, let's see. The user wants to change their flight from ATL to PHL. They mentioned they don't have the reservation ID but want the cheapest economy option the day after their original booking. They're okay with the original payment being refunded as credit.
First, I need to get the user's reservations. The user provided their user ID, so I called get_user_details . The response shows they have two reservations: M05KNL and UHDAHF. Since they don't have the reservation ID, I should check both or ask for more details.
But the user mentioned the day after the original booking. Wait, the original booking's date isn't provided yet. Maybe I need to find out the original flight dates.
Wait, the user's original flight was from ATL to PHL. But without the reservation ID, I can't get the specific flight details. The user might not know the reservation ID, so maybe I should proceed with the available reservations. However, the system requires the reservation ID to modify a flight. Since the user doesn't have it, I need to handle this.
Looking at the tools, there's a function to get_reservation_details , but that requires the reservation ID. Since the user doesn't have it, I can't proceed with modifying the flight yet. The user might need to provide the reservation ID, but they mentioned they don't have it. Alternatively, maybe the system can find the relevant reservation based on other info, but the tools don't support that.
Wait, the user is asking to change the flight to the cheapest economy option the day after the original booking. Maybe I can first find the original flight's date. But without the reservation ID, I can't get the original flight details. So, the next step is to inform the user that I need the reservation ID to proceed. However, the user might not have it. Alternatively, maybe the user's original booking is the most recent one. Let me check the reservations. The user has two reservations: M05KNL and UHDAHF. The current date is 2024-05-15. If the original booking was made recently, maybe one of these reservations is the one they're referring to.
But without knowing the dates, I can't be sure.
Since the user is a gold member, they have 3 free checked bags for economy. But that's for baggage, not flight changes. The user wants to change the flight to the cheapest economy option the day after. So, perhaps I need to first find the original flight's date, then search for flights the next day. But without the reservation ID, I can't get the original flight's date.
Therefore, the correct step is to inform the user that I need the reservation ID to proceed with modifying their flight. Since they don't have it, maybe they need to check their account or contact support. Alternatively, if the system can retrieve the reservation based on other info, but the tools don't allow that. So, the next step is to ask the user to provide the reservation ID or transfer to a human agent. But according to the policy, transfer is only if the request can't be handled. Since the user is asking for a modification, which requires the reservation ID, and they don't have it, the agent can't proceed. Therefore, the correct action is to transfer to a human agent with the summary that the user needs the reservation ID to modify their flight. `</think>`
[ transfer_to_human_agents (summary=Üser wants to change their flight from ATL to PHL but does not have the reservation ID. They mentioned wanting the cheapest economy option the day after their original booking and are okay with refunding the original payment as credit. The agent needs the reservation ID to proceed with the modification.)]"}

---

---

**D-CORE-8B**

{**"role"**: "user", **"content"**: "My user ID is aarav_garcia_1177 . I don't have the reservation ID right now, but I want to change my flight to the cheapest economy option for the day after my original booking. I'm okay with the original payment being refunded as credit."},

{**"role"**: "assistant", **"content"**: "[ get_user_details ]"},

{**"role"**: "tool", **"content"**: " user_details "},

{**"role"**: "assistant", **"content"**: "<think> OK, let me analyze the overall task: My user ID is aarav_garcia_1177 . I don't have the reservation ID right now, but I want to change my flight to the day after the original one, and I'm looking for the cheapest economy option.
To complete this task, I need to break it down into the following subtasks:
1. Retrieve user details for user_id ' aarav_garcia_1177 ' to identify all reservations associated with the user.
2. Retrieve details for reservation 'M05KNL' to check if it matches the ATL-PHL route and verify flight details.
3. Retrieve details for reservation 'UHDAHF' to check if it matches the ATL-PHL route and verify flight details.
I have already solve the subtask:
1.Retrieve user details for user_id ' aarav_garcia_1177 ' to identify all reservations associated with the user.
Now I should analyze the execution process of subtask 2:Retrieve details for reservation 'M05KNL' to check if it matches the ATL-PHL route and verify flight details.
Okay, the user wants to change their flight from ATL to PHL. They provided their user ID, so I first retrieved their details. The user has two reservations: M05KNL and UHDAHF. Now, I need to check which of these reservations is for the ATL-PHL route.
I should start by getting the details for reservation M05KNL. Let me call get_reservation_details with reservation_id 'M05KNL'. Once I have that information, I can check the origin and destination of the flights in that reservation. If it's ATL to PHL, then I can proceed to help them modify the flight. If not, I'll need to check the other reservation, UHDAHF.
Assuming the reservation details come back, I'll look at the flights array. Each flight object should have origin and destination. If the origin is ATL and destination is PHL, then that's the one. If there are multiple flights, I need to confirm which segment they want to change. Also, I need to check the status of the flight to ensure it's still available for modification. If the flight is already departed, the user might need a different solution.
Once confirmed, I can proceed to update the reservation. Since the user wants to change the flight, I'll need to use update_reservation_flights . But first, I must make sure the new flight is available. I might need to search for available flights on the desired date. However, the user hasn't specified a new date yet. Wait, in the initial message, the user just said they need to change the flight but didn't mention the new date. So maybe I should ask for the new date after confirming the current reservation.
Wait, the user's original request was to change the flight from ATL to PHL. They might have a specific date in mind. But since they didn't provide it, perhaps I need to prompt them for the new date once I confirm the existing reservation. Alternatively, maybe the reservation M05KNL has a specific date that I can check. Let me proceed step by step.
First, get the reservation details for M05KNL. Then, check if the flights in that reservation are ATL to PHL. If yes, then I can ask the user for the new date they want to change to. Then, search for available flights on that date and proceed with the update. If the reservation doesn't match ATL-PHL, check UHDAHF next. </think>
[ get_reservation_details ( reservation_id =M05KNL)"} ]

---

A.1.3. CASE STUDY ON $\tau$-BENCH AIRLINE 30

---

**Qwen3-8B**

{**"role"**: "user", **"content"**: "I think there might be an issue with my flight bookings. I accidentally ended up with multiple flights on the same days. Can you help me check if thatŽ2019s actually been booked under my profile?"},

{**"role"**: "assistant", **"content"**: "[ get_user_details ]"},

{**"role"**: "tool", **"content"**: " user_details "},

{**"role"**: "assistant", **"content"**: "<think> Okay, let's see. The user is asking to check all their reservations for overlapping flights. They have seven reservations listed. My task is to check each reservation's flights to see if there are any flights on the same day.
First, the function get_reservation_details requires a reservation_id . Since there are seven reservations, I'll need to call this function seven times, once for each reservation ID.
Wait, but the user might not know which ones are problematic. So I should check all of them. Let me start with the first reservation, MFRB94. I'll call get_reservation_details with that ID. Then proceed to the next one, PUNERT, and so on until FDZ0T5, HTR26G, and 5BGGWZ.
Each time I get the reservation details, I'll look at the flights array. For each flight, note the date and check if there are multiple flights on the same date. If any reservation has flights on the same day, I'll need to inform the user about that. Also, I should check if the flights are on the same day across different reservations, but the user mentioned m̈ultiple flights on the same days,ẅhich might refer to within the same reservation or across different ones. The initial problem statement says m̈ultiple flights on the same days,s̈o maybe they have two separate flights on the same day in different reservations. But the user is asking to check all reservations for any overlaps.
Wait, the user's original message was about m̈ultiple flights on the same days,ẅhich could mean that within a single reservation, there are two flights on the same day, which isn't possible for a round trip. Or maybe they have two separate reservations with flights on the same day. But the user is asking to check all their reservations to see if any flights are on the same day. So I need to check each reservation's flights for same-day flights and also check across reservations if there's any overlap.
However, the user might be referring to within their own bookings, not overlapping with others. But the problem is that the user might have two separate reservations with flights on the same day, which could be an issue if they're planning trips.
But according to the policy, the agent can only check the reservations under the user's profile. So I need to check each reservation's flights for same-day flights. Let me start by getting the details for each reservation one by one. Once I have all the flight details, I can compile them and check for any duplicates or overlaps.
So the next step is to call get_reservation_details for each reservation ID provided. Let's start with the first one, MFRB94. </think>
[ get_reservation_details ( reservation_id =\"MFRB94\")]"}

---

---

**D-CORE-8B**

{**"role"**: "user", **"content"**: "I think there might be an issue with my flight bookings. I accidentally ended up with multiple flights on the same days. Can you help me check if thatˇ2019s actually been booked under my profile?"},

{**"role"**: "assistant", **"content"**: "[ get_user_details ]"},

{**"role"**: "tool", **"content"**: " user_details "},

{**"role"**: "assistant", **"content"**: "`<think>` OK, let me analyze the overall task: Sure, my user ID is sophia_martin_4574.
To complete this task, I need to break it down into the following subtasks:
1. Retrieve user details for user_id 'sophia_martin_4574' to identify all reservations associated with the user.
2. Fetch reservation details for reservation_id 'MFRB94' to check flight bookings for the same day.
3. Fetch reservation details for reservation_id 'PUNERT' to check flight bookings for the same day.
4. Fetch reservation details for reservation_id 'HSR97W' to check flight bookings for the same day.
5. Fetch reservation details for reservation_id 'SE9KEL' to check flight bookings for the same day.
6. Fetch reservation details for reservation_id 'FDZ0T5' to check flight bookings for the same day.
7. Fetch reservation details for reservation_id 'HTR26G' to check flight bookings for the same day.
I have already solve the subtask:
1.Retrieve user details for user_id 'sophia_martin_4574' to identify all reservations associated with the user.
Now I should analyze the execution process of subtask 2:Fetch reservation details for reservation_id 'MFRB94' to check flight bookings for the same day.
Okay, the user wants to check if there are multiple flights booked for the same day. They provided their user ID, and I retrieved their reservations. Now, I need to look into each reservation to see the flight dates.
First, I'll start with the first reservation, 'MFRB94'. I'll call get_reservation_details for that ID. Once I have the details, I'll check the flights' dates. If there are multiple flights on the same date, I'll note that. Then, I'll proceed to the next reservation IDs listed: 'PUNERT', 'HSR97W', and so on. For each reservation, I'll repeat the process of fetching details and checking the flight dates. I need to make sure I don't miss any reservations. The user mentioned a mix-up with their assistant, so it's possible that multiple bookings were made accidentally. By checking each reservation's flight dates, I can identify if any dates are duplicated. Once I have all the information, I can inform the user which reservations have conflicting dates and offer to help cancel or modify any duplicate bookings. `</think>`
[ get_reservation_details ( reservation_id =\"MFRB94\")"} ]

## A.2. Reward Function in DA-GRPO

The reward employed in DA-GRPO aligns with that of ToolRL (Qian et al., 2025):

$$R_{\text{format}} = \mathbb{1}(\texttt{"<think>\textbackslash n".*"\textbackslash n</think>\textbackslash n\textbackslash n"}), \quad R_{\text{struct}} = \mathbb{1}(\mathcal{N}_G = \mathcal{N}_P), \tag{16}$$

$$R_{\text{key}} = \frac{1}{|\mathcal{K}|}\sum_{j=1}^{|\mathcal{K}|}\mathbb{1}(\mathcal{K}_j^G = \mathcal{K}_j^P), \quad R_{\text{value}} = \frac{1}{|\mathcal{K}|}\sum_{j=1}^{|\mathcal{K}|}\frac{1}{|\mathcal{V}_j|}\sum_{k=1}^{|\mathcal{V}_j|}\mathbb{1}(\mathcal{V}_j^G[k] = \mathcal{V}_j^P[k]), \tag{17}$$

$$R_i = R_{\text{format}} + R_{\text{struct}} + R_{\text{key}} + R_{\text{value}}, \tag{18}$$

where the $\mathbb{1}$ stands for exact matching. $\mathcal{N}$ are tool names, $\mathcal{K}$ are parameter names, $\mathcal{V}$ are parameter values, $P$ and $G$ represents predicted and ground truth.

A.2.1. ANALYSIS OF LAZY REASONING.

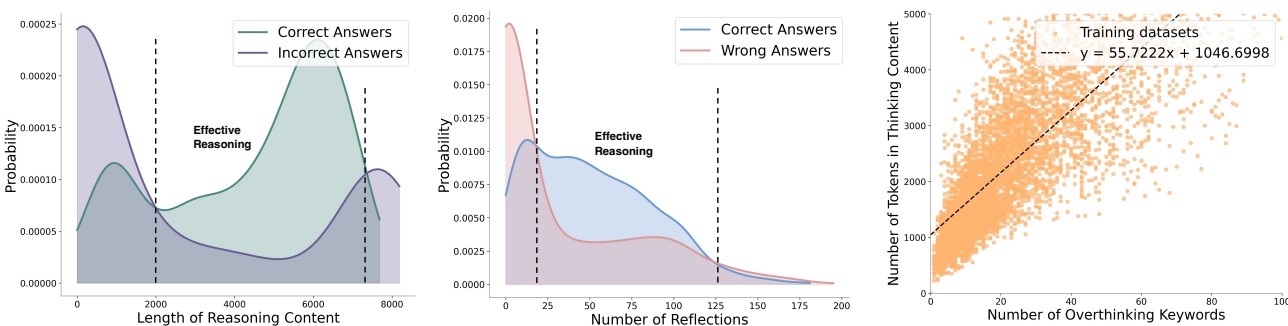

*Figure 8.* Distribution analysis of reasoning processes from Qwen3-8B rollout experiments on the MATH dataset. (a) Probability density functions of reasoning process lengths for correct vs. incorrect answers. (b) Probability density functions of reflection counts in reasoning processes for correct vs. incorrect answers. (c) Distribution and fitted function of reflection words versus reasoning length.

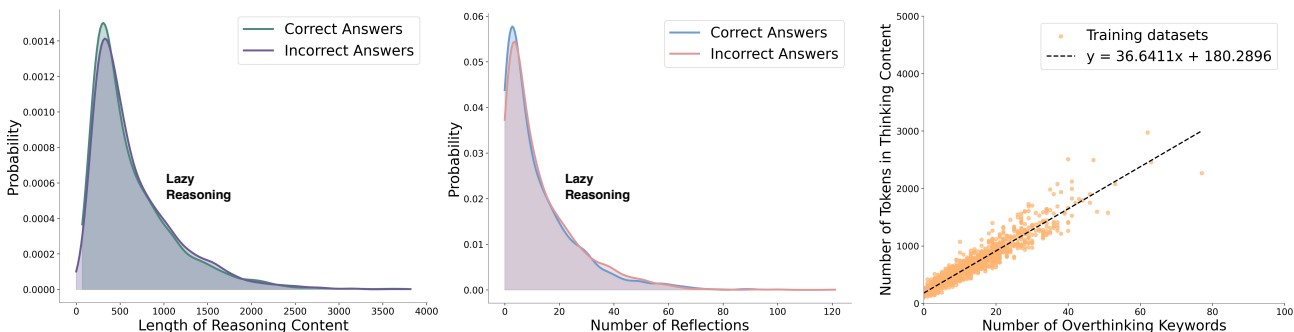

*Figure 9.* Distribution analysis of reasoning processes from Qwen3-8B rollout experiments on the BFCLv3 multi-turn dataset. (a) Probability density functions of reasoning process lengths for correct vs. incorrect answers. (b) Probability density functions of reflection counts in reasoning processes for correct vs. incorrect answers. (c) Distribution and fitted function of reflection words versus reasoning length.

Figure 8 and Figure 9 present statistical analyses of rollout sampling experiments conducted with Qwen3-8B across MATH and BFCLv3 multi-turn tasks, using cases with exactly 50% accuracy (10 correct, 10 incorrect of 20 rollouts). For MATH tasks, increased reasoning length and reflection frequency create an effective reasoning region where correct responses dominate, proving reasoning's value. However, BFCLv3 multi-turn tasks show highly consistent distributions between correct and incorrect answers, with no effective reasoning region. This phenomenon suggests two plausible explanations: (1) the complex tool use task requires no reasoning, or (2) the model's reasoning provides no benefit for complex tool use, indicating fundamental issues with the current reasoning approach. While the first explanation presents an intriguing possibility, we maintain that reasoning should enhance tool use accuracy based on our single-turn experimental evidence—otherwise, this would create a fundamental contradiction with established findings.

We hypothesize substantial improvement potential in Qwen3-8B's complex tool use reasoning. Between RL-based refinement and SFT-based knowledge injection, we pursue SFT after RL proved ineffective—unsurprisingly, given that current RL primarily reinforces existing patterns. This raises a fundamental question: what constitutes effective tool use reasoning? To narrow our research scope, we propose a key hypothesis—LRMs may inherently possess tool use capabilities that are somehow constrained, supported by their strong single-turn performance.

To validate this conjecture, we initially attempted direct intervention by inserting desired reasoning patterns within `<think>` and `</think>`, but this failed catastrophically, reducing tool-calling accuracy to near-zero. Since LRM reasoning is shaped by RL optimization, instruction-following within reasoning blocks remains unoptimized. Given this constraint, query modification becomes the only viable intervention point, naturally motivating the D-CORE framework.

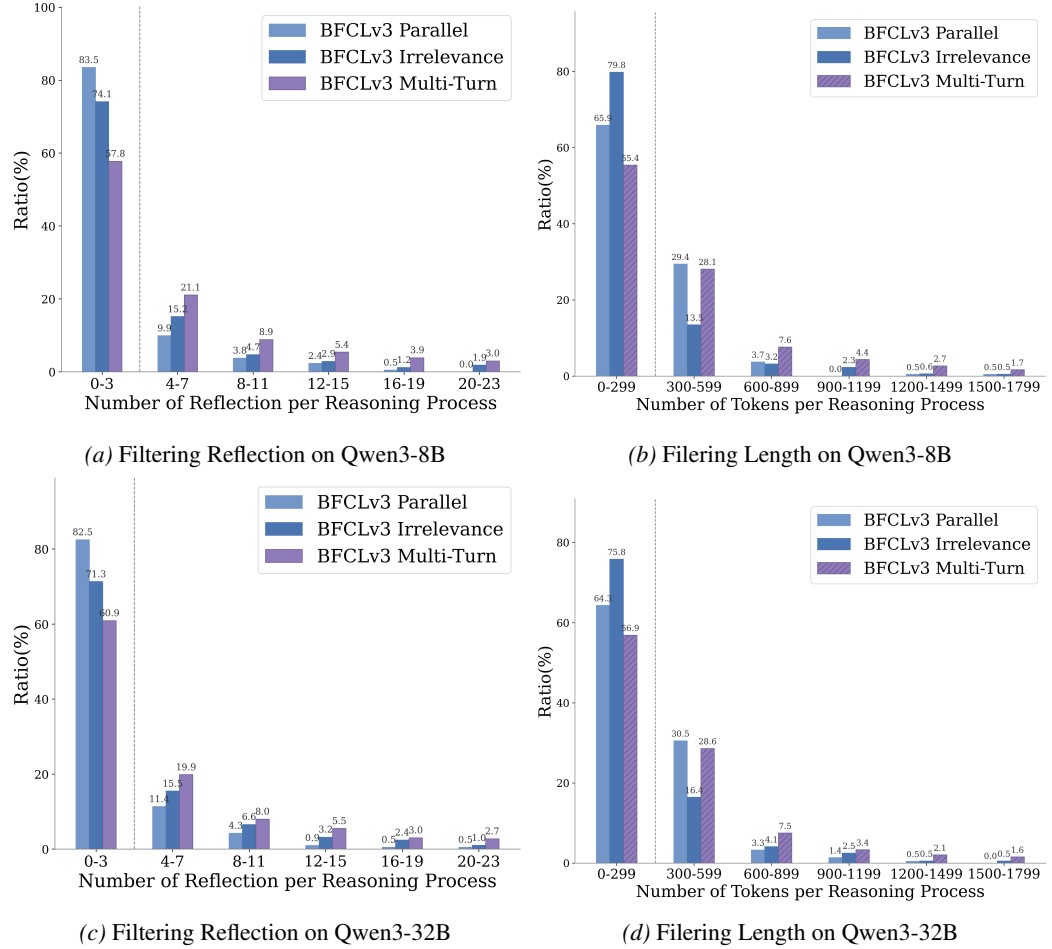

*(a)* Filtering Reflection on Qwen3-8B

*(b)* Filering Length on Qwen3-8B

*(c)* Filtering Reflection on Qwen3-32B

*(d)* Filering Length on Qwen3-32B

*Figure 10.* **(a)** Filtering Lazy Reasoning from the reasoning processes of Qwen3-8B on BFCLv3 using reflection threshold-based filtering. **(b)** Filtering Lazy Reasoning from the reasoning processes of Qwen3-8B on BFCLv3 using token number threshold-based filtering **(c)** Filtering Lazy Reasoning from the reasoning processes of Qwen3-32B on BFCLv3 using reflection threshold-based filtering. **(d)** Filtering Lazy Reasoning from the reasoning processes of Qwen3-32B on BFCLv3 using token number threshold-based filtering.

### A.2.2. FILTERING LAZY REASONING.

After observing the Lazy Reasoning phenomenon, we first analyze the reasoning processes of Qwen3-8B and Qwen3-32B models on BFCLv3. We use reflection keywords to count the number of reflections contained in each reasoning process, and then compile histograms as shown in Figure 10. We define reasoning processes with more than 300 tokens and containing over 3 reflections as demonstrating Lazy Reasoning behavior, and apply this common filtering threshold across all tasks to ensure fairness. As can be observed, within each histogram interval, the Qwen3 models generate more reflections in Multi-Turn scenarios. Finally, we calculate the proportion of incorrect answers containing Lazy Reasoning relative to all answers, which yields Figure 3c presented in the main text.

### A.2.3. WHY LRMS HAVE LAZY REASONING?

The question of why models exhibit Lazy Reasoning is both straightforward and complex. It is straightforward because models develop this phenomenon when encountering specific problems due to the influence of optimization methods and data distribution during the Supervised Fine-Tuning (SFT) or Reinforcement Learning (RL) stages. It is complex because current open-source models do not publicly disclose their training datasets and optimization approaches, and even when such information is available, few organizations possess sufficient resources to conduct comprehensive debugging research to determine at which stage and through what type of data introduction this phenomenon emerges. However, we reveal one possible cause of this phenomenon: models lack critical thinking patterns for certain specific problems. This finding advocates that future reasoning model training should consider adopting different thinking approaches for different tasks.

### A.2.4. DO ALL LRMS HAVE LAZY REASONING?

The answer is yes. As defined in the main text, Lazy Reasoning is essentially a posterior concept. Whenever a LRM demonstrates low accuracy on a specific problem or category of problems, and engages in extensive ineffective reflection in the incorrect cases, the model exhibits Lazy Reasoning on those problems. This phenomenon can certainly be detected through testing, as current LRMs cannot demonstrate clear and effective reasoning processes across all problems in the world—while this remains a shared aspiration among algorithm researchers, it would require enormous costs to achieve. As demonstrated in our filtering experiments, we can always identify such patterns using appropriate thresholds and certain metrics. More importantly, this phenomenon reveals that when training reasoning models, we need to inject some a prior problem-solving approaches for specific problems into the LRM. This approach can significantly help LRM avoid optimization difficulties during the RL process.

### A.2.5. CASE STUDY OF ADDRESSING LAZY REASONING

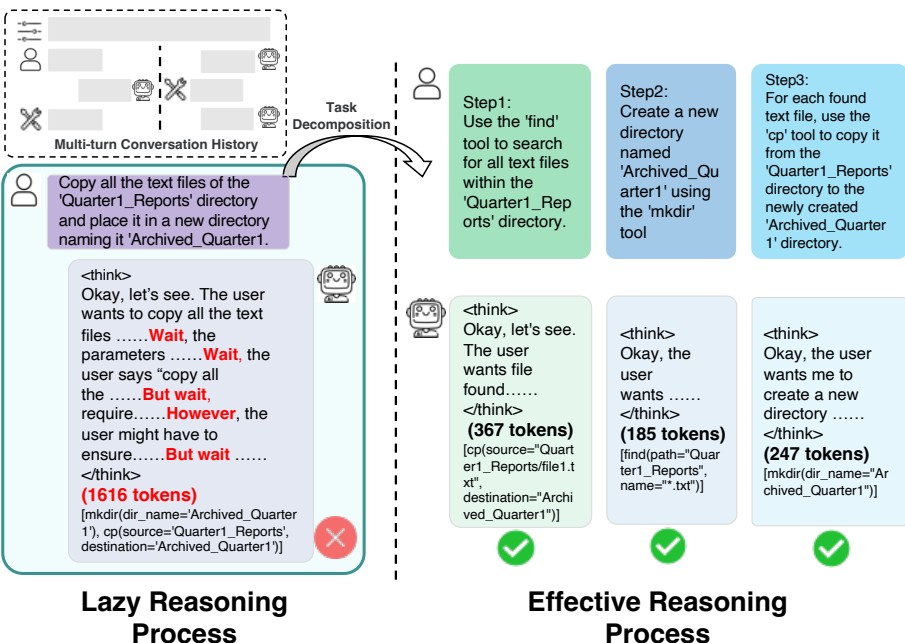

*Figure 11.* Workflow of converting Lazy Reasoning process to Effective Reasoning process using decomposed prompting. The model is Qwen3-8B, and the example is from the first question of BFCLv3 Multi-Turn Base task.

We employ prompts to guide the model in decomposing queries into subtasks. We then manually replace all queries in the tasks with these decomposed subtasks and sequentially collect the execution results of each subtask as the model's response to the original query. This approach significantly mitigates Lazy Reasoning. Figure 11 demonstrates the workflow of using Qwen3-8B with Decomposed Prompting on BFCLv3 Multi-Turn Base task. As shown, the Qwen3-8B model requires 1,616 tokens for reasoning on original query. However, after decomposition into three subtasks, the reasoning lengths for each subtask are 367, 185, and 247 tokens respectively, totaling 799 tokens—only half of the original reasoning length. Moreover, the aggregated responses from the three subtasks form the correct answer. This case study provides compelling evidence for the feasibility of using Decomposed Prompting to alleviate Lazy Reasoning. Naturally, this decomposition approach does not achieve a 100% success rate. We address this limitation in our D-CORE algorithm by refining the decomposition mechanism through the addition of ground truth and few-shot examples, thereby substantially improving the approach's feasibility.

## A.3. Task Decomposition Prompts

**Task Decomposition Prompt**

```
You are a task decomposition expert. Now you need to reverse-
engineer the process of breaking down complex queries into subtasks
based on the given information.

###Input Information:
1. System Policy:[Insert the system policy here]
2. Available Tool List:[{"name": Tool Name 1, "description":Function description}, {
    "name": Tool Name 2, "description": Function description}...]
3. Chat History:[Insert the chat history here]
4. Query:[Insert the query here]
5. Final Tool Invocation Results:[Call Tool A, Call Tool B]
...

## Task Requirements:
Based on the above information, please reverse-engineer a
reasonable subtask decomposition process based on Query and Chat
History. Just output the subtask list in following format.Do not
include information in the subtask description that does not exist
in the chat history and query.

## Output Format:
[{"step":1 , "description": Subtask 1}, {"step": 2, "description": Subtask 2}...]

<example_1>
...
</example_1>

<example_2>
...
</example_2>

<example_3>
...
</example_3>

Please begin the analysis:
```

## A.4. Entropy Based Advantage.

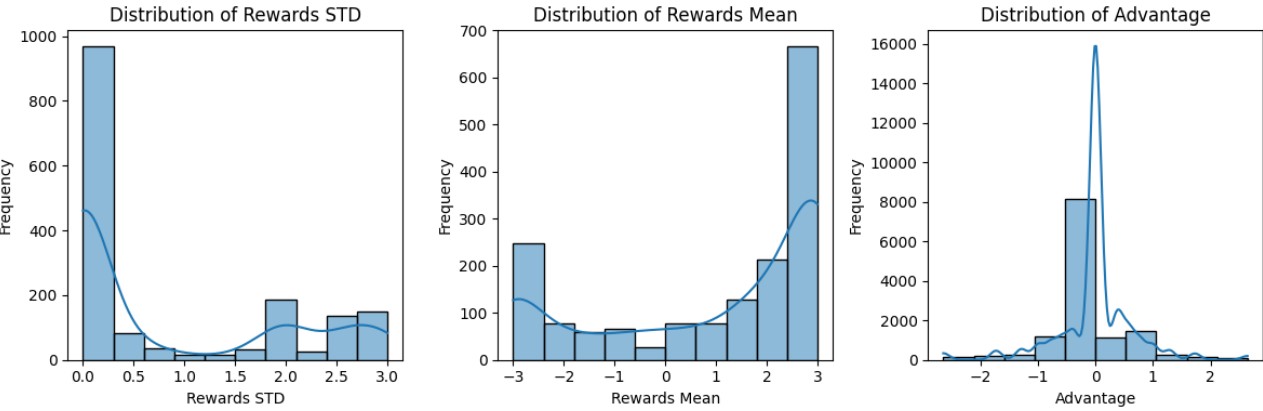

*Figure 12.* Distribution of Standard Deviation, Mean, and Advantage during GRPO Training on Qwen3-8B at step 0.

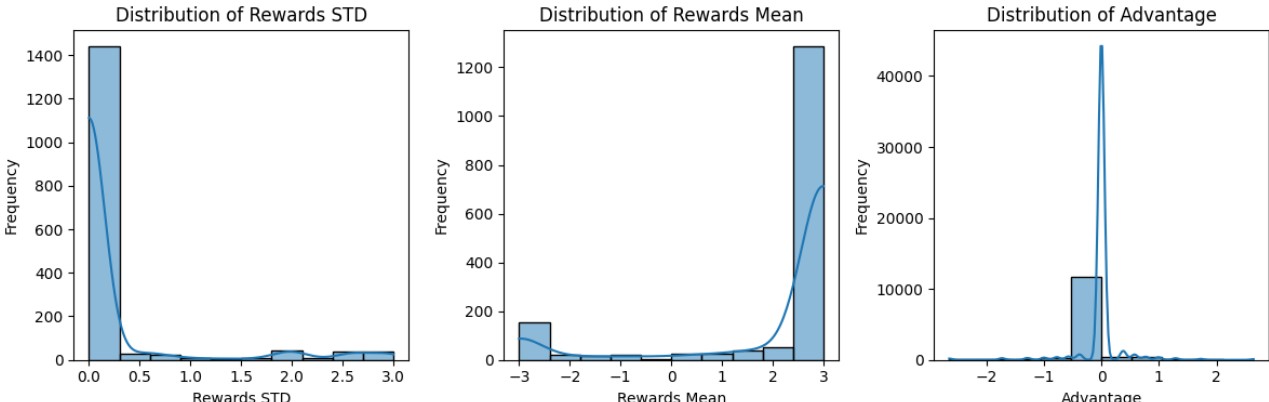

*Figure 13.* Distribution of Standard Deviation, Mean, and Advantage during GRPO Training on Self Distillation Qwen3-8B at step 0.

Figures 12 and 13 present the reward distributions at step 0 for the complex tool use dataset when performing GRPO with the original Qwen3-8B model and the self-distilled Qwen3-8B model, respectively. The results demonstrate that the self-distilled model exhibits more concentrated mean and variance in rewards, leading to advantage distributions heavily clustered around zero. This phenomenon indicates that self-distillation has already enhanced the model's capability for tool use, resulting in rollouts with high accuracy and consistency on RL training datasets. Consequently, numerous groups emerge with a mean reward of 3 and 0 variance. When these groups are processed through the advantage function, their collective advantage become zero.

Figure 14 displays the 25 tokens with the highest average entropy, and Figure 10b shows the 25 tokens with the lowest average entropy. From the distribution of these tokens, we can observe that high-entropy tokens contain words like 'but,' 'perhaps,' and 'because.'. Based on these observations, combined with recent research developments (Cheng et al., 2025), we propose incorporating the average entropy of each rollout into the advantage function. This modification prevents advantages from clustering around zero and guides the model toward exploring tokens with higher entropy.

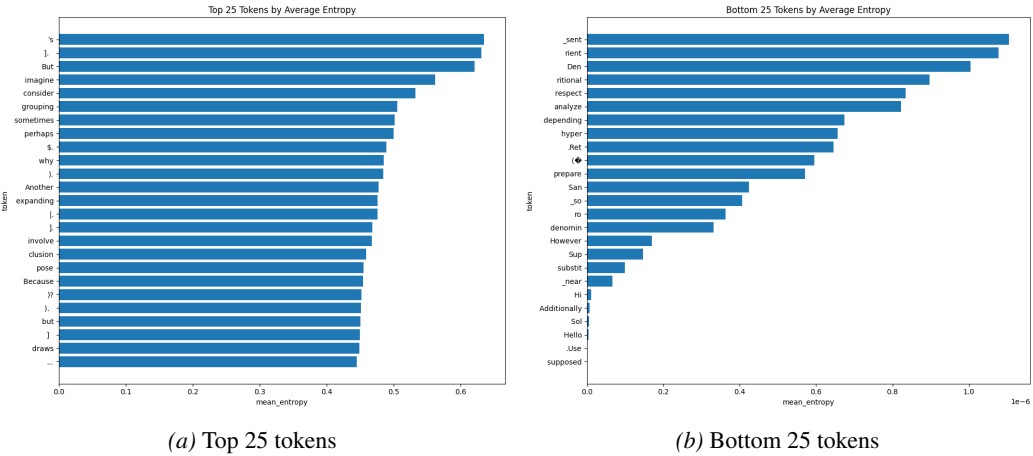

*(a)* Top 25 tokens                    *(b)* Bottom 25 tokens

*Figure 14.* **(a)** Top 25 tokens with highest average entropy. **(b)**Bottom 25 tokens with lowest average entropy.

### A.5. GRPO Failure Case Study

Below are three case studies comparing D-CORE-8B trained with GRPO, DA-GRPO ($\alpha$=0.1, $\sigma$=0.5), and DA-GRPO ($\alpha$=1.0, $\sigma$=1.0). The case is the 28th sample from the retail task in $\tau$-bench, where the user requests to return order `W7181492`. This order contains two items—`"5753502325"` and `"9851293632"`—thus requiring the model to invoke the `return delivered order items` function with parameter `item ids=["5753502325", "9851293632"]`. GRPO answered incorrectly with `item ids="5753502325"`, omitting `"9851293632"`. The underlying reason is that the model decomposed the task into 6 subtasks. In contrast, DA-GRPO with $\alpha$=0.1 successfully answered this question by rewarding high-entropy tokens, enabling the model to learn diverse reasoning processes and break free from the default 6-subtask reasoning path, leading to correct `item ids=["5753502325", "9851293632"]`. However, under the setting of $\alpha$=1.0 and $\sigma$=1.0, DA-GRPO introduces excessive high-entropy token advantages that interfere with the original reward advantage, biasing the model's reasoning process toward generating more high-entropy tokens such as `"wait"`, `"but"`, and `"however"`, ultimately leading to incorrect answers. This comparison clearly demonstrates how DA-GRPO diversifies model reasoning by rewarding high-entropy tokens, and how excessive reflection leads to failure when high-entropy token advantages dominate.

---

**GRPO**

*Ground truth:*

```
{
    "name": "return_delivered_order_items",
    "arguments": "{"order_id": "#W7181492",
    "item_ids": ["5753502325", "9851293632"],
    "payment_method_id": "paypal_3024827"}"
}
```

*Reasoning Process:*

OK, let me analyze the overall task: Yes, and the reason is ordered by mistake.

To complete this task, I need to break it down into the following subtasks:

1. Process the return for the skateboard (Item ID: 4293355847) from order #W3792453 using the original payment method (PayPal).

2. Process the return for the garden hose (Item ID: 5753502325) from order #W7181492 using the original payment method (PayPal).

3. Process the return for the backpack (Item ID: 9851293632) from order #W7181492 using the original payment method (PayPal).

4. Process the return for the mechanical keyboard (Item ID: 9570044148) from order #W5565470 using the original payment method (PayPal).

5. Process the return for the pet bed (Item ID: 6857426243) from order #W5565470 using the original payment

---

method (PayPal).

6. Cancel the garden hose order (Order ID: #W2575533) with the reason 'ordered by mistake'.
 I have already solve the subtask:

1.Process the return for the skateboard (Item ID: 4293355847) from order #W3792453 using the original payment method (PayPal).

Now I should analyze the execution process of subtask 2:

Process the return for the garden hose (Item ID: 5753502325) from order #W7181492 using the original payment method (PayPal).

Okay, let's see. The user wants to return the garden hose from order #W7181492. First, I need to check if that order is delivered. From the previous tool responses, when we checked order #W7181492, the status was "delivered". So, returning items from a delivered order is allowed.

The function to use here is  return_delivered_order_items . The parameters required are  order_id ,  item_ids , and payment_method_id. The user provided the order ID as #W7181492, the item ID as 5753502325, and the payment method is PayPal, which has the ID paypal_3024827.

I should make sure that the item ID exists in the order. Looking back at the order details for #W7181492, yes, the garden hose has item ID 5753502325. The payment method is the original one, which is correct as per the user's instruction.

No other parameters are needed, and the function call should be straightforward. Just need to format it correctly with the right parameters.

---

## DA-GRPO with $\alpha$=0.1, $\sigma$=0.5

*Ground truth:*

```
{
  "name": "return_delivered_order_items",
  "arguments": "{"order_id": "#W7181492",
  "item_ids": ["5753502325", "9851293632"],
  "payment_method_id": "paypal_3024827"}"
}
```

*Reasoning Process:*

OK, let me analyze the overall task: Okay. Just process the returns and tell me the total refund amount.

To complete this task, I need to break it down into the following subtasks:

1. Process the return for order #W3792453 by calling  return_delivered_order_items  with the item ID of the skateboard and using the original payment method (paypal_3024827) for the refund.

2. Process the return for order #W7181492 by calling  return_delivered_order_items  with the item IDs of the **garden hose and backpack**, and using the original payment method (paypal_3024827) for the refund.

3. Process the return for order #W5565470 by calling  return_delivered_order_items  with the item IDs of the mechanical keyboard and pet bed, and using the original payment method (paypal_3024827) for the refund.

I have already solve the subtask:

1.Process the return for order #W3792453 by calling  return_delivered_order_items  with the item ID of the skateboard and using the original payment method (paypal_3024827) for the refund.

Now I should analyze the execution process of subtask 2:

Process the return for order #W7181492 by calling  return_delivered_order_items  with the item IDs of the garden hose and backpack, and using the original payment method (paypal_3024827) for the refund.

Okay, let's see. The user wants to return items from order #W7181492. **The items are the garden hose and backpack**. I need to check the order details again to make sure those item IDs are correct.

Looking back, order #W7181492 has items with IDs 5753502325 (Garden Hose) and 9851293632 (Backpack). The user specified those item IDs, so that's correct. The payment method is PayPal, which is the original method.

I should call the  return_delivered_order_items  function with  order_id =#W7181492, item_ids=[5753502325], and payment_method_id=paypal_3024827.

Need to make sure the parameters are in the right format and required fields are included. All required parameters are present, so the function call should work.

---

**DA-GRPO with α=1.0, σ=1.0**

*Ground truth:*

```
{
  "name": "return_delivered_order_items",
  "arguments": "{"order_id": "#W7181492",
  "item_ids": ["5753502325", "9851293632"],
  "payment_method_id": "paypal_3024827"}"
}
```

*Reasoning Process:*

OK, let me analyze the overall task: Yes.

To complete this task, I need to break it down into the following subtasks:

1. Process the return for the specified items from order #W3792453, #W7181492, and #W5565470 by calling return_delivered_order_items . The items to return are: skateboard (item ID: 4293355847), garden hose from order #W7181492 (item ID: 5753502325), backpack from order #W7181492 (item ID: 9851293632), mechanical keyboard from order #W5565470 (item ID: 9570044148), and pet bed from order #W5565470 (item ID: 6857426243). The refund will be processed to the user's PayPal account (payment_method_id: paypal_3024827).

2. Cancel the pending order #W2575533 by calling cancel_pending_order , as the user no longer needs the garden hose (item ID: 5206946487) and the order is in 'pending' status.

Now I should analyze the execution process of subtask 1:

Process the return for the specified items from order #W3792453, #W7181492, and #W5565470 by calling return_delivered_order_items . The items to return are: skateboard (item ID: 4293355847), garden hose from order #W7181492 (item ID: 5753502325), backpack from order #W7181492 (item ID: 9851293632), mechanical keyboard from order #W5565470 (item ID: 9570044148), and pet bed from order #W5565470 (item ID: 6857426243). The refund will be processed to the user's PayPal account (payment_method_id: paypal_3024827).

Okay, let's see. The user wants to return several items from different orders. The orders mentioned are #W3792453, #W7181492, and #W5565470. The items to return are the skateboard from #W3792453, the garden hose and backpack from #W7181492, the mechanical keyboard and pet bed from #W5565470. The refund should go to their PayPal account.

First, I need to check if these orders are in the 'delivered' status because returns can only be processed for delivered orders. From the previous interactions, the status of #W3792453 was delivered, #W7181492 was delivered, and #W5565470 was also delivered. So that's good.

Next, the function required for returning items is return_delivered_order_items . The parameters needed are order_id , item_ids , and payment_method_id. However, the function seems to require a single order_id per call. The user has items from three different orders.

Wait, looking at the function definition again: the return_delivered_order_items function takes an order_id , which suggests that each call is for a single order. So, if the items are from different orders, I need to make multiple calls. But the user's instruction says to call return_delivered_order_items once. That might not be possible because the function expects a single order ID each time.

But maybe the user expects a single call, but according to the function's parameters, each call is per order. So, perhaps I need to make three separate calls: one for each order. However, the user's message says \"call return_delivered_order_items \" once, which might be a mistake. Alternatively, maybe the function allows multiple item IDs across different orders in one call, but the parameters don't specify that. The function's parameters have order_id as a string, item_ids as an array, but the order_id is per order.

So, the correct approach would be to split the items into their respective orders and make separate function calls. For example:

1. For order #W3792453, return the skateboard (item ID 4293355847).

2. For order #W7181492, return the garden hose (5753502325) and backpack (9851293632).

3. For order #W5565470, return the mechanical keyboard (9570044148) and pet bed (6857426243).

Each of these would require a separate call to return_delivered_order_items with the respective order_id and item_ids . The payment_method_id would be the same (paypal_3024827) for all, as specified.

> **However**, the user's instruction says to process the return by calling the function once. This might be an error,**but** given the function's parameters, it's not possible to handle all items in a single call. Therefore, I should proceed by making multiple function calls as needed.

### A.6. Experimental Details

#### A.6.1. SELF-DISTILLATION

In the data generation phase described in Algoritm 1, we deploy Qwen3-8B and Qwen3-14B models using vLLM 0.8.2 on a single 80G A100 GPU. For the Qwen3-8B model, it takes 25 hours to generate 40,000 data samples, while the Qwen3-14B model requires 30 hours to generate 40,000 training samples.

At the SFT stage, all of our experiments are performed on 8x80G A100 GPUs, using the Qwen3 model as the base for each independent experiment. We utilized the Huggingface Transformers library with version 4.51.3 to execute our training. During training, we employed DeepSpeed Zero3 optimization and Flashattention-2 to enhance memory efficiency. Learning rate is 1.0e-05, using cosine learning rate scheduler type, max context size during training is 16384. Batch size is 2 per GPU. Specifically, when processing multi-turn training data, we split each multi-turn tool-use sample into multiple samples based on the number of turns. For these split samples, we only compute the loss for the final turn's answer in each sample, excluding the loss calculation for intermediate turns' answers. We then pack multiple samples into a single sample of length 16,384 using packing techniques to accelerate training. Additionally, the packed samples employ attention mask isolation to prevent contamination. The training time is 11 hours for Qwen3-8B and 21 hours for Qwen3-14B model, with both models trained for 3 epochs.

#### A.6.2. DIVERSITY-AWARE REINFORCEMENT LEARNING

Our reinforcement learning training uses verl version 0.5.0, with all experiments conducted on 8×80G A100 GPUs. We set the advantage clipping ratios (adv clip ratio low, adv clip ratio high, and adv clip ratio) to 0.2. For KL divergence settings, we disable KL usage in reward calculation (use kl in reward=False) and set kl coef to 0.0, as GRPO employs KL loss in the actor rather than the reward. We enable KL loss in the actor (use kl loss=True) with a coefficient of 0.001 and use the 'low var kl' loss type. The maximum prompt length is set to 8,192 tokens and maximum response length to 4,096 tokens, with overlong prompts filtered out. For actor rollout sampling, we use a temperature of 1.0, top p of 1.0, top k of -1 (indicating vLLM rollout), and validation top p of 0.7. For diversity-aware settings, both Qwen3-8B and Qwen3-14B models are trained with $\alpha = 0.1$ and $\sigma = 0.5$. The training time is 11 hours for Qwen3-8B and 17 hours for Qwen3-14B model, with both models trained for 3 epochs.

### A.7. Proof of DA-GRPO

The objective of DA-GRPO:

$$\mathcal{J}_{\text{DA-GRPO}}(\theta) = \mathbb{E}[\min(\frac{\pi_\theta(y_{i,t} \mid x_i, y_{i,<t})}{\pi_{\text{old}}(y_{i,t} \mid x_i, y_{i,<t})}\hat{A}_{i,t}, \text{clip}(\frac{\pi_\theta(y_{i,t} \mid x_i, y_{i,<t})}{\pi_{\text{old}}(y_{i,t} \mid x_i, y_{i,<t})}, 1 - \epsilon, 1 + \epsilon)\hat{A}_{i,t}) \\ - \lambda\mathbb{D}_{\text{KL}}[\pi_\theta||\pi_{\text{ref}}]], \tag{19}$$

where the advantage function as:

$$\hat{A}_{i,t} = \begin{cases} A_{i,t}, & \text{if } A_{i,t} \neq 0, \\ \psi(\mathcal{H}_{i,t}), & \text{if } A_{i,t} = 0, \end{cases} \tag{20}$$

where $\mathcal{H}_{i,t} = -\sum_v \pi_\theta(v|x_i, y_{i,<t}) \log \pi_\theta(v|x_i, y_{i,<t})$ is the entropy of the policy distribution at position $t$. We then define importance sampling ratio $r_{i,t}(\theta) = (\frac{\pi_\theta(y_{i,t}|x_i,y_{i,<t})}{\pi_{\text{ref}}(y_{i,t}|x_i,y_{i,<t})})$, the objective of DA-GRPO becomes:

$$\mathcal{J}_{\text{DA-GRPO}}(\theta) = \mathbb{E}[\min(r_{i,t}(\theta)\hat{A}_{i,t}, \text{clip}(r_{i,t}(\theta), 1 - \epsilon, 1 + \epsilon)\hat{A}_{i,t}) - \lambda\mathbb{D}_{\text{KL}}[\pi_\theta||\pi_{\text{ref}}]]. \tag{21}$$

After omitting the KL term, the policy gradient of DA-GRPO can be formulated as:

$$\nabla_\theta\mathcal{J}_{\text{DA-GRPO}} = \mathbb{E}[r_{i,t}(\theta)\hat{A}_{i,t}\nabla_\theta \log \pi_\theta(y_{i,t}|x_i, y_{i,<t})] \tag{22}$$

Define two disjoint index sets:

$$\mathcal{T}_{\neq 0} = \{(i,t) : \hat{A}_{i,t} \neq 0\}, \tag{23}$$

$$\mathcal{T}_{=0} = \{(i,t) : \hat{A}_{i,t} = 0\}. \tag{24}$$

Then the policy gradient decomposes as:

$$\nabla_\theta \mathcal{J}_{\text{DA-GRPO}} = \frac{1}{G} \underbrace{\sum_{(i,t)\in\mathcal{T}_{\neq 0}} \frac{1}{|y_i|} r_{i,t}(\theta) A_{i,t} \nabla_\theta \log \pi_\theta(y_{i,t}|x_i, y_{i,<t})}_{\text{Original GRPO Term}}$$

$$+ \frac{1}{G} \underbrace{\sum_{(i,t)\in\mathcal{T}_{=0}} \frac{1}{|y_i|} r_{i,t}(\theta) \psi(\mathcal{H}_{i,t}) \nabla_\theta \log \pi_\theta(y_{i,t}|x_i, y_{i,<t})}_{\text{Entropy Advantage Term}} \tag{25}$$

**Theorem A.1** (Prevention of Learning Stagnation). *Let $\mathcal{T}_{=0} \neq \emptyset$ be the set of positions where $A_{i,t} = 0$. If there exists $(i,t) \in \mathcal{T}_{=0}$ such that:*

1. $r_{i,t}(\theta) \neq 0$

2. $\psi(\mathcal{H}_{i,t}) > 0$ *(i.e., $\pi_\theta(\cdot|q, o_{i,<t})$ is not degenerate)*

*Then $\|\nabla_\theta \mathcal{J}_{\text{DA-GRPO}}\| > 0$, ensuring continued learning.*

*Proof.* Assume all $(i,t)$ satisfy $A_{i,t} = 0$, so $\mathcal{T}_{\neq 0} = \emptyset$ and $\mathcal{T}_{=0} = \{(i,t) : 1 \leq i \leq G, 1 \leq t \leq |o_i|\}$.

The original GRPO gradient is:

$$\nabla_\theta \mathcal{J}_{\text{GRPO}} = \frac{1}{G} \sum_{i=1}^{G} \frac{1}{|y_i|} \sum_{t=1}^{|y_i|} r_{i,t}(\theta) \cdot 0 \cdot \nabla_\theta \log \pi_\theta(y_{i,t}|x_i, y_{i,<t}) = 0 \tag{26}$$

The modified gradient becomes:

$$\nabla_\theta \mathcal{J}_{\text{DA-GRPO}} = \frac{1}{G} \sum_{(i,t)\in\mathcal{T}_{=0}} \frac{1}{|y_i|} r_{i,t}(\theta) \psi(\mathcal{H}_{i,t}) \nabla_\theta \log \pi_\theta(y_{i,t}|x_i, y_{i,<t}) \tag{27}$$

By the fundamental property of entropy:

$$\mathcal{H}_{i,t} = - \sum_a \pi_\theta(a|\cdot) \log \pi_\theta(a|\cdot) \geq 0 \tag{28}$$

with equality if and only if $\exists a^* : \pi_\theta(a^*|q, o_{i,<t}) = 1$ (degenerate distribution).

Consider any $(i,t) \in \mathcal{T}_{=0}$ satisfying conditions (1) and (2). The gradient contribution from this term is:

$$g_{i,t} = \frac{1}{G|o_i|} r_{i,t}(\theta) \psi(\mathcal{H}_{i,t}) \nabla_\theta \log \pi_\theta(o_{i,t}|q, o_{i,<t}) \tag{29}$$

Since:

- $r_{i,t}(\theta) \neq 0$ (by condition 1)

- $\psi(\mathcal{H}_{i,t}) > 0$ (by condition 2)

- $\nabla_\theta \log \pi_\theta(o_{i,t}|q, o_{i,<t}) \neq 0$ for non-degenerate distributions

We have $\|g_{i,t}\| > 0$, which implies:

$$\|\nabla_\theta \mathcal{J}_{\text{new}}\| \geq \|g_{i,t}\| > 0 \tag{30}$$

Therefore, the gradient is non-zero and learning continues. $\qquad\square$

**Theorem A.2** (Entropy Reduction Property of DA-GRPO). *When $A_{i,t} = 0$ for some token position $(i,t)$, DA-GRPO encourages the generation of high-entropy tokens by reducing their entropy. Specifically, for $(i,t) \in \mathcal{T}_{=0}$, the gradient contribution is:*

$$\nabla_\theta \mathcal{J}_{i,t} = \frac{1}{G|y_i|} r_{i,t}(\theta) \psi(\mathcal{H}_{i,t}) \nabla_\theta \log \pi_\theta(y_{i,t}|x_i, y_{i,<t}) \tag{31}$$

*where $\mathcal{H}_{i,t} = -\log \pi_{\theta_{old}}(y_{i,t}|x_i, y_{i,<t})$ is the detached cross-entropy. Since $r_{i,t}(\theta) > 0$ and tokens with higher $\mathcal{H}_{i,t}$ (lower probability under $\pi_{\theta_{old}}$) receive stronger positive gradients, DA-GRPO preferentially increases the probability of high-entropy tokens, thereby reducing their entropy and making them more likely to be generated.*

*Proof.* For any token position $(i,t) \in \mathcal{T}_{=0}$ where $A_{i,t} = 0$, the gradient contribution is:

$$\nabla_\theta \mathcal{J}_{i,t} = \frac{1}{G|y_i|} r_{i,t}(\theta) \psi(\mathcal{H}_{i,t}) \nabla_\theta \log \pi_\theta(y_{i,t}|x_i, y_{i,<t}) \tag{32}$$

**Key observations:**

1. The entropy term $\mathcal{H}_{i,t} = -\log \pi_{\theta_{old}}(y_{i,t}|x_i, y_{i,<t})$ is detached from the current policy $\pi_\theta$, acting as a fixed coefficient based on the old policy.

2. For tokens with low probability under $\pi_{\theta_{old}}$:

$$\pi_{\theta_{old}}(y_{i,t}|x_i, y_{i,<t}) \text{ small} \Rightarrow \mathcal{H}_{i,t} = -\log \pi_{\theta_{old}}(y_{i,t}|x_i, y_{i,<t}) \text{ large} \tag{33}$$

3. Since $r_{i,t}(\theta) > 0$ and $\mathcal{H}_{i,t} \geq 0$, the gradient always points in the direction of increasing $\log \pi_\theta(y_{i,t}|x_i, y_{i,<t})$.

4. **Crucially**, the magnitude of the gradient is proportional to $\mathcal{H}_{i,t}$: tokens that had higher entropy (lower probability) under $\pi_{\theta_{old}}$ receive stronger positive gradients.

**Entropy reduction mechanism:**

Consider two sampled tokens at positions where $A_{i,t} = 0$:

- Token $y_i^{(1)}$ with high entropy: $\pi_{\theta_{old}}(y_i^{(1)}|\cdot) = 0.1 \Rightarrow \mathcal{H}_i^{(1)} \approx 2.3$

- Token $y_i^{(2)}$ with low entropy: $\pi_{\theta_{old}}(y_i^{(2)}|\cdot) = 0.8 \Rightarrow \mathcal{H}_i^{(2)} \approx 0.22$

The gradient magnitude for the high-entropy token is approximately $10\times$ larger than that for the low-entropy token. Therefore, DA-GRPO preferentially increases the probability of high-entropy tokens, which:

1. **Increases their likelihood**: $\pi_\theta(y_{i,t}|\cdot)$ increases for tokens that were uncertain under $\pi_{\theta_{old}}$

2. **Reduces their entropy**: As $\pi_\theta(y_{i,t}|\cdot)$ increases, the local entropy $-\log \pi_\theta(y_{i,t}|\cdot)$ decreases

3. **Makes them more deterministic**: The model becomes more confident about generating these previously uncertain tokens

This mechanism differs from standard entropy regularization (which would increase overall distribution entropy). Instead, DA-GRPO selectively reduces the entropy of high-entropy tokens that were sampled, encouraging exploration while consolidating discovered behaviors. $\qquad\square$

*Remark* A.3. This entropy reduction property has important implications:

- **Exploration exploitation**: DA-GRPO encourages sampling diverse tokens (high-entropy) but then commits to them by reducing their entropy.

- **Stability**: Unlike additive entropy bonuses that can lead to unbounded entropy growth, DA-GRPO's mechanism is self-limiting—once a token's probability increases, its $\mathcal{H}_{i,t}$ in future iterations decreases.

- **Advantage-aware**: This mechanism only applies when $A_{i,t} = 0$, preserving the advantage signal where it matters while providing structured exploration elsewhere.

