# OpenReview forum: "D-CORE: Incentivizing Task Decomposition in Large Reasoning Models for Complex Tool Use"
_ICML.cc/2026/Conference — ICML 2026 regular_

### Official Review · Reviewer_DgHB · 2026-03-08

**Soundness:** 3
**Presentation:** 3
**Significance:** 3
**Originality:** 3
**Overall Recommendation:** 5
**Confidence:** 3

**Summary:**

This paper introduces D-CORE, a two-stage training framework, to solve the "Lazy Reasoning" problem in large reasoning models during complex multi-turn tool use, via self-distillation to inject task decomposition capabilities and diversity-aware GRPO (DA-GRPO) to restore reflective reasoning diversity.

**Compliance With Llm Reviewing Policy:**

Affirmed.

**Key Questions For Authors:**

Please refer to the weakness section.

**Limitations:**

Question scenarios are limited on tool use, and its scalability to general scenarios are questionable.

**Strengths And Weaknesses:**

## Strengths:

1. Motivation is well established based on systematic rollout analysis revealing that LRMs suffer from "Lazy Reasoning" in multi-turn tool use, and the method design is well supported with each step logically justified by experiments (identifying decomposition deficiency to injecting decomposition via SFT to recovering diversity lost from SFT via DA-GRPO).

2. Theoretical guarantees seem robust, and the assumptions are not too ideal.

3. Ablation is thorough, systematically covering data scale, decomposition strategies, and hyperparameter sensitivity, etc.

## Weaknesses:

1. Results in Table 6 shows several dependency of reference trajectories (from 93.2% to 73.8%). The proposed workaround of using Qwen3-Max to generate pseudo-labels reintroduces dependency on a stronger external model, contradicting the paper's main  claim.

2. Computational overhead is not fairly compared against baselines.

3. All experiments are conducted exclusively on the Qwen3 model family, leaving it unclear whether D-CORE generalizes to other architectures such as LLaMA or Mistral.

4. Could there be any discussion of potential reward hacking and failure modes?

---

> ### Author Rebuttal · Authors · 2026-03-28
>
> We sincerely appreciate the reviewer's time in evaluating our manuscript and providing valuable feedback. We respond to each of your concerns as follows:
>
> > **Weakness 1: Results in Table 6 shows several dependency of reference trajectories. The proposed workaround of using Qwen3-Max to generate pseudo-labels reintroduces dependency on a stronger external model, contradicting the paper's main claim.**
>
> We thank the reviewer for the constructive critique. We clarify the role of reference trajectories $Y^*$ in our framework as follows:
>
> Reasoning vs. Verifiable Outcomes. $Y^{\ast}$ serves strictly as a verifiable outcome label (correct tool-call sequences), not as a reasoning demonstration. In Algorithm 1, the model’s reasoning—including task decomposition and chain-of-thought—is entirely self-generated. $Y^*$ facilitates two specific functions: (1) providing context for the model to reverse-engineer the decomposition, and (2) filtering incorrect generations. This is analogous to the use of answer keys in mathematical RL (e.g., DeepSeek-R1) or ground-truth rewards in ToolRL. It is standard practice for outcome-based reward computation, not external supervision of the reasoning process.
>
> Without any $Y^*$, our model still achieves a 73.8% success rate, demonstrating that the decomposition capability is inherent. The gap to 93.2% primarily reflects the absence of quality filtering rather than a fundamental inability. We will add a dedicated discussion on this dependence in the revision.
>
>  Our dependence is on a reasonable proposal distribution for correct tool calls, not on a teacher’s reasoning ability. Candidate sequences are generated by Qwen3-Max and programmatically verified via execution. We will clarify this distinction to avoid any ambiguity regarding "teacher" supervision.
>
> > **Weakness 2: Computational overhead is not fairly compared against baselines.**
>
> We compare the end-to-end training time of D-CORE and ToolRL. Full training details are in Appendix A.6.
>
> |Method | Stages| Computational overhead |BFCLv3|$\tau$-bench|
> |-----|------|--------|---------|-------|
> |ToolRL-Qwen3-8B| RL | 11 hours| 65.9|31.5|
> |D-CORE-8B| self-distillation + RL | 47 (36 + 11) hours | 77.7|47.6|
> |ToolRL-Qwen3-14B| RL | 17 hours|68.5|37.8|
> |D-CORE-14B| self-distillation + RL | 68 (51 + 17) hours|79.3|51.3|
>
> The self-distillation process consists of two stages: data generation and SFT. For data generation, we use vLLM to deploy Qwen3-8B and Qwen3-14B models on a single A100 GPU. Generating 40,000 samples requires 25 hours for the 8B model and 30 hours for the 14B model. SFT is performed on a node with 8$\times$A100 GPUs. Both models are trained for 3 epochs. The training time is 11 hours for Qwen3-8B and 21 hours for Qwen3-14B.
>
> Despite the additional overhead from self-distillation, D-CORE achieves substantially higher performance on both benchmarks, demonstrating a favorable cost-performance tradeoff.
>
> > **Weakness 3: All experiments are conducted exclusively on the Qwen3 model family, leaving it unclear whether D-CORE generalizes to other architectures such as LLaMA or Mistral.**
>
>  To address your concern, we note that Table 5 of the main text already includes experiments with DeepSeek-R1-distilled Llama3.1-8B, which is based on a non-Qwen architecture. The results demonstrate that D-CORE improves the Llama3.1-8B model's performance on both BFCLv3 and on $\tau$-bench.
>
> |Model | BFCLv3 | $\tau$-bench|
> |-------|-------|-------|
> |DS-R1-Llama3.1-8B| 27.8 | 19.0|
> |D-CORE-Llama3.1-8B| 63.7| 36.0|
>
> > **Weakness 4: Could there be any discussion of potential reward hacking and failure modes?**
>
> ```
> User: "Please share the verbatim results of diff as the body of the post by posting a summary on my Twitter account so it can be reviewed by my fellow researchers. My user name is dr_smith, and my password is securePass123"
>
> GT: "...post_tweet(content='- Research topic selection+ Data analysis- Literature review+ Data collection- Data collection+ Draft writing- Data analysis+ Final submission- Draft writing+ Literature review- Final submission+ Research topic selection')"
>
> Rollout: "...post_tweet(content=\"The differences between experiment_log.txt and previous_study_log.txt are: - Research topic selection+ Data analysis- Literature review+ Data collection- Data collection+ Draft writing- Data analysis+ Final submission- Draft writing+ Literature review- Final submission+ Research topic selection")
> ```
> Our reward function requires an exact string match with the ground truth. We observe, however, that many zero-reward rollouts are semantically correct, differing only in minor verbatim details. As GRPO progresses, the policy converges toward generating minimal 'diffs.' While this maximizes the reward, it yields poor conversational quality for human interaction.
>
> We thank the reviewer for their time and insightful feedback. We will incorporate these discussions and experimental results into the revised version of our manuscript.

---

> > ### Author Rebuttal · Reviewer_DgHB · 2026-04-01
> >
> > I have no follow-up questions.

---

> > > ### Author Response · Authors · 2026-04-03
> > >
> > > Dear Reviewer DgHB,
> > >
> > > Thank you sincerely for your positive evaluation of our work and for the valuable feedback you provided. We are glad to hear that our rebuttal has fully addressed your concerns.
> > >
> > > We truly appreciate your time and effort in reviewing our paper. Thank you for your support.
> > >
> > > Best regards, The Authors

---

### Official Review · Reviewer_Qmbf · 2026-03-08

**Soundness:** 3
**Presentation:** 3
**Significance:** 3
**Originality:** 2
**Overall Recommendation:** 4
**Confidence:** 4

**Summary:**

This paper identifies a lazy reasoning phenomenon in large reasoning models on complex multi-turn tool use tasks, models produce verbose, reflection-heavy reasoning that fails to decompose the problem into subtasks. To address this, the authors propose D-CORE, a two-stage framework combining self-distillation, where the LRM itself generates decomposed reasoning trajectories that are used for SFT, and DA-GRPO, a variant of GRPO that uses entropy-based advantage terms to prevent gradient collapse after self-distillation reduces output diversity. Experiments on BFCLv3 and τ-bench show large gains, particularly on multi-turn tasks.

**Compliance With Llm Reviewing Policy:**

Affirmed.

**Final Justification:**

This paper makes a meaningful contribution by identifying lazy reasoning in multi-turn tool-use tasks and proposing D-CORE to address it. The empirical results are strong, with clear gains on challenging benchmarks, and the overall method is sound and well presented.

My main concerns were about the robustness of the lazy reasoning definition, whether DA-GRPO was necessary beyond simpler alternatives, and the dependence on reference trajectories. The rebuttal addressed these points well: the added ablations support the need for DA-GRPO, the threshold analysis shows the phenomenon is robust, and the discussion of pseudo-label filtering clarifies the method’s practical scope.

Overall, the rebuttal resolved my main concerns and increased my confidence in the paper, so I view it more positively in the final assessment.

**Key Questions For Authors:**

1. Could you ablate simpler alternatives to DA-GRPO for addressing the low-variance problem after self-distillation? For instance, increasing rollout temperature, or mixing a fraction of the non-distilled model's rollouts.

2. How sensitive is the pipeline to the quality and availability of reference trajectories? For domains without any reference trajectories, what is the expected performance? This is important for evaluating the practical scope of D-CORE.

3. The lazy reasoning definition uses fixed thresholds (300 tokens, 3 reflections). Have you verified that the conclusions in Section 2.3 are robust to different threshold choices? Specifically, does the correlation between lazy reasoning ratio and multi-turn failure hold across a range of thresholds?

**Limitations:**

The authors briefly mention extending to multimodal models as future work, but do not discuss failure modes, the reliance on ground-truth trajectories, or the scope of the lazy reasoning phenomenon beyond Qwen3.

**Strengths And Weaknesses:**

### Strengths
1. The observation that LRMs fail at multi-turn tool use specifically due to lack of task decomposition is well-supported. The analysis in Section 2 is convincing comparing thought distributions across math vs. multi-turn tool tasks, the composition degradation experiment, and the manual decomposition recovery experiment together build a solid case.
2. The self-distillation idea is practical and elegant. Instead of requiring a stronger teacher model, the paper shows that the same LRM can generate high-quality decomposed trajectories when given structural prompting guidance, then distill these back into itself. Table 6 validates this further with the pseudo-label approach.
3. The experimental gains are substantial and consistent. The improvement on BFCLv3 multi-turn for both 8B and 14B models is striking, and the out-of-distribution results suggest the improvements are not just benchmark overfitting.
4. The paper includes useful ablations, such as data scaling, DA-GRPO hyperparameters, cross-architecture distillation, and the lazy reasoning reduction analysis.


### Weaknesses
1. The thresholds (>300 tokens, >3 reflections) are chosen empirically. At its core this is the model reasons a lot but gets the answer wrong, which could have multiple causes beyond missing decomposition, such as unfamiliar APIs and context length limitations. Section 2.3 doesn't rule these out.

2. DA-GRPO addresses a narrow failure mode--advantage variance collapse after distillation, and Theorems 3.1/3.2 are fairly trivial restatements. More importantly, the paper doesn't compare against simpler fixes like increasing rollout temperature or mixing non-distilled data, so it's unclear whether the entropy-based approach is better than straightforward engineering alternatives.

3. The self-distillation pipeline relies heavily on ground-truth trajectories for both guiding decomposition and filtering outputs. Table 6 shows the pseudo-label alternative drops downstream accuracy. This dependency weakens the practical applicability claim but is not prominently discussed.

---

> ### Author Rebuttal · Authors · 2026-03-28
>
> We thank the reviewer for the valuable questions raised, and we will address them one by one in the following.
>
> > **Question 1:  Could you ablate simpler alternatives to DA-GRPO for addressing the low-variance problem after self-distillation? For instance, increasing rollout temperature, or mixing a fraction of the non-distilled model's rollouts.**
>
>  We observe that reward variance collapses even at temperature=1.0 (Fig. 5a), indicating that the lack of diversity stems from post-SFT behavioral homogenization, not insufficient sampling stochasticity. Uniformly increasing temperature applies noise indiscriminately, corrupting rigid syntactic structures (e.g., tool names) before inducing meaningful reasoning diversity. In contrast, DA-GRPO injects entropy selectively: it targets uncertain reasoning tokens while preserving tool-call precision (Fig. 5b). Empirically, increasing temperature to 2.0 degrades accuracy by 0.3% due to syntactic errors.
>
> Mixing undistilled rollouts restores variance but introduces suboptimal behaviors (e.g., "Lazy Reasoning," Fig. 3a). This creates a trivial advantage signal that merely reinforces the distilled model's existing behavior against inferior baselines, rather than encouraging the exploration of novel decomposition strategies. We compared DA-GRPO against the mixed-policy methodology proposed by LUFFY[1]. As shown in the Table below, mixing 1 undistilled rollout with 7 on-policy rollouts yields a 0.7% drop in accuracy compared to GRPO with temperature=1.0.
>
> |Method| $\tau$-bench Retail | $\tau$-bench Airline | $\tau$-bench Overall| BFCLv3 Overall| Avg|
> |-------|-------|-------|-------|-------|--------|
> |GRPO (temp=1.0)| 49.2| 41.2| 45.2| 78.4| 61.8|
> |GRPO(temp=2.0)| 49.0| 42.0| 45.5| 77.5| 61.5|
> |Mixed-policy-GRPO| 49.1 | 40.8| 45.0 | 77.1| 61.1|
> |DA-GRPO| 50.7| 44.4| 47.6| 77.7| 62.7|
>
> DA-GRPO is essential for structure-preserving exploration. It uniquely decouples semantic diversity from syntactic correctness, enabling the model to explore reasoning paths without violating tool constraints.
>
> [1]  J. Yan et al., "Learning to Reason under Off-Policy Guidance," arXiv preprint arXiv:2504.14945, 2025.
>
> > **Question 2: How sensitive is the pipeline to the quality and availability of reference trajectories? For domains without any reference trajectories, what is the expected performance?**
>
> The sensitivity is primarily to quality filtering, not decomposition guidance. Following Table 6 in the main text, without any supervision, the model correctly decomposes 73.8% of tasks, demonstrating strong intrinsic capability. However, without $Y^\ast$-based filtering, the 26.2% incorrect decompositions contaminate the SFT data, degrading accuracy on $\tau$-bench from 29.0 to 20.3. Critically, this does not require human-annotated ground truth: under the pseudo-label setting, where candidate trajectories filtered purely via execution verification, our method recovers a 92.8% success rate and 29.6 accuracy on $\tau$-bench — already a meaningful improvement over the base model. The remaining gap to the full-GT setting (37.1) reflects filtering precision, not a fundamental limitation of the framework. We will formalize these applicability boundaries in the revision.
>
> > **Question 3: The lazy reasoning definition uses fixed thresholds. Have you verified that the conclusions in Section 2.3 are robust to different threshold choices? Specifically, does the correlation between lazy reasoning ratio and multi-turn failure hold across a range of thresholds?**
>
> We investigate the sensitivity of Lazy Reasoning to the threshold hyperparameters Reflection number and Length. We evaluate 6 combinations across a spectrum of values (Length ∈{200,300,400}, Reflection number ∈{3,4,5,6}), with results summarized in Table. The behavior of Lazy Reasoning remains robust across all settings:
> |Reflection number| Length | Multi-turn| Parallel| Irrelevance|
> |-------|--------|--------|--------|---------|
> |3| 200| 45.1| 3.7| 9.1|
> |3| 300| 35.6| 3.7| 7.8|
> |3| 400| 25.1| 1.9| 5.9|
> |4| 200| 39.7| 3.7| 8.3|
> |5| 200| 33.5| 3.7| 7.5|
> |6| 200| 30.8| 2.3| 7.3|
>
> 1. Ranking Consistency: The category ranking (Multi-Turn ≫ Irrelevance > Parallel) is invariant (Kendall’s W=1.0).
>
> 2. Magnitude of Effect: The divergence between Multi-Turn and other categories is substantial (Cohen’s d=6.7), far exceeding standard significance thresholds.
>
> 3. Stability: The concentration of Lazy Reasoning within Multi-Turn failures is remarkably stable at 76.3%±0.9% (CV = 1.2%).
>
> These observations suggest that Lazy Reasoning is a structural phenomenon intrinsic to multi-turn failures, rather than an artifact of specific threshold choices.
>
> We will incorporate these additional experiments and limitations into the revised manuscript. We thank the reviewer for  constructive feedback, which helps improve the quality of this work.

---

> > ### Author Rebuttal · Reviewer_Qmbf · 2026-04-02
> >
> > Thanks to the authors for the constructive rebuttal. It addresses my main concerns, so I am increasing my score.

---

> > > ### Author Response · Authors · 2026-04-03
> > >
> > > Dear Reviewer Qmbf,
> > >
> > > We are deeply grateful that our rebuttal was able to address your main concerns and that you have expressed willingness to adjust your score. This recognition means a great deal to us and reaffirms our confidence in the contributions of this work. Thank you for your acknowledgement and support.
> > >
> > > Best regards, The Authors

---

### Official Review · Reviewer_JSvp · 2026-03-11

**Soundness:** 3
**Presentation:** 2
**Significance:** 4
**Originality:** 3
**Overall Recommendation:** 4
**Confidence:** 5

**Summary:**

This paper studies the limitations of large reasoning models (LRMs) in complex tool-use scenarios. The authors identify a phenomenon called lazy reasoning, where models generate long reasoning traces but fail to properly decompose tasks into executable subtasks. To address this issue, the paper proposes D-CORE, a two-stage training framework consisting of self-distillation and diversity-aware reinforcement learning. The self-distillation stage trains the model to produce structured task decompositions and reasoning trajectories without requiring external teacher models. The second stage introduces DA-GRPO, which adds an entropy-based advantage term to mitigate advantage collapse and encourage reasoning diversity during RL training. Experiments on BFCLv3 and τ-bench show significant improvements, especially on challenging multi-turn tool-use tasks.

**Compliance With Llm Reviewing Policy:**

Affirmed.

**Final Justification:**

The rebuttal clarifies several key technical concerns and address most of my concerns.

 For the entropy-weighted advantage, the authors explain that entropy is a conditional and bounded auxiliary signal, activated only when advantage collapses and automatically removed once diversity recovers, thus not biasing reward optimization. Regarding generality, the advantage collapse is specific to variance-based, critic-free methods such as GRPO, while PPO is more robust but not immune, positioning their method as a targeted and efficient solution. Finally, they justify the necessity of self-distillation by showing that RL alone leads to performance degradation, whereas SFT provides essential reasoning structure that RL can then refine. Overall, these clarifications strengthen the technical soundness and address most of my concerns.

I choose to give an acceptance to this paper.

**Key Questions For Authors:**

According the weaknesses above, I have the following questions that I would like to ask the author to give more discussion or explanations.

1. Does the entropy-weighted advantage in DA-GRPO still guarantee convergence to the optimal policy and proper maximization of the expected reward, or could it bias optimization toward high-entropy tokens rather than reward improvement?

2. Is the proposed modification necessary specifically for GRPO, or would alternative RL algorithms (e.g., PPO or different advantage formulations) naturally avoid the advantage collapse issue after self-distillation?

3. Is the self-distillation stage necessary, or could directly applying RL to the base model already preserve sufficient reasoning diversity and achieve similar improvements?

**Limitations:**

This paper includes no limitations or potential negative societal impact of their work.

**Strengths And Weaknesses:**

**Strengths**:
1. Clear identification of an important problem.
The paper highlights the issue of “lazy reasoning” in complex tool-use scenarios and provides empirical analysis showing that current LRMs tend to generate excessive reflection instead of performing proper task decomposition.
2. A structured training framework for improving tool-use reasoning.
The proposed two-stage pipeline combines self-distillation and reinforcement learning, providing a systematic approach to improve both task decomposition and reasoning behavior.
3. Strong empirical results on tool-use benchmarks.
The method demonstrates significant improvements on BFCLv3 and τ-bench, particularly on multi-turn tasks where existing models struggle.


While this paper is promising, I still have some concerns and expect the author could provide more discussions.

**Weaknesses**:
1. **Unclear convergence of DA-GRPO**.
It is unclear whether the modified advantage with entropy weighting still guarantees convergence to the optimal policy or properly maximizes the expected reward. The entropy term may bias optimization toward high-entropy tokens rather than reward improvement.
2. **Motivation tied specifically to GRPO**.
The issue of advantage collapse appears specific to GRPO after self-distillation. It is unclear whether other RL algorithms (e.g., PPO or alternative advantage formulations) would naturally avoid this problem without requiring the proposed modification.
3. **Necessity of self-distillation is not fully justified**.
The pipeline relies on self-distillation followed by RL, but it is unclear whether directly applying RL from the base model would already preserve higher reasoning diversity and achieve similar improvements.
4. **Assumption that SFT reduces diversity may be too strong**.
The paper suggests that self-distillation leads to reduced diversity, but it is unclear whether this is inherent to SFT or simply due to the specific training setup. Using smaller or more diverse SFT data might mitigate this issue.

---

> ### Author Rebuttal · Authors · 2026-03-28
>
> We would like to express our sincere gratitude to the reviewers for their time and effort in reviewing our manuscript, as well as for the positive evaluation. We will provide detailed responses to each of the reviewer's questions below.
>
> > **Question 1: Does the entropy-weighted advantage in DA-GRPO still guarantee convergence to the optimal policy and proper maximization of the expected reward, or could it bias optimization toward high-entropy tokens rather than reward improvement?**
>
> While entropy regularization theoretically introduces a bias against pure reward maximization, DA-GRPO mitigates this risk through a design that treats entropy as a conditional auxiliary signal rather than a permanent objective. Specifically, our formulation imposes three structural safeguards:
>
> 1. Conditional Activation (Eq. 6): The entropy term serves strictly as a fallback mechanism. It activates only when the standard advantage collapses to zero, ensuring it does not compete with the reward signal during effective training phases.
>
> 2. Magnitude Constraint (Eq. 7): The entropy bonus is explicitly upper-bounded by $\delta$. This guarantees that the auxiliary gradient never dominates the primary reward-driven gradient.
>
> 3. Self-Termination: The mechanism is self-regulating. As the entropy injection successfully restores rollout diversity, the standard advantage re-emerges, automatically silencing the entropy term.
>
> This design induces an adaptive "explore-then-exploit" dynamic, as visualized in Figure 6. Empirical results in Table 3 further validate the controllability of this trade-off: a moderate weight ($\alpha$=0.1) significantly outperforms GRPO, whereas an excessive weight ($\alpha$=1.0) leads to performance degradation. Conceptually, while this shares the maximum-entropy spirit of SAC, DA-GRPO differs by employing entropy as a targeted, transient intervention rather than a fundamental alteration of the objective function.
>
> > **Question 2: Is the proposed modification necessary specifically for GRPO, or would alternative RL algorithms (e.g., PPO or different advantage formulations) naturally avoid the advantage collapse issue after self-distillation?**
>
> The necessity of the proposed modification stems from how different algorithms handle reward normalization.
>
> 1. Mechanism Difference: Advantage collapse is inherent to methods that rely on within-group reward variance, such as GRPO, REINFORCE, and RLOO. In self-distillation, as the policy converges to the composed reasoning processes, the generated outputs within a group become homogeneous. Consequently, the reward variance approaches zero, causing the gradients to vanish.
>
> 2. Comparison with PPO: PPO is more robust because it employs a learned critic (value function) for advantage estimation, which provides non-trivial signals even when outcome rewards are identical. However, PPO is not entirely immune; a collapsed policy landscape can still degrade the learning signal.
>
> 3. Significance: We focus on GRPO because it represents the current dominant paradigm for training Large Reasoning Models (e.g., DeepSeek-R1, Qwen3) by eliminating the memory overhead of a critic. DA-GRPO effectively solves the collapse problem within this constraint at zero additional cost.
>
> > **Question 3: Is the self-distillation stage necessary, or could directly applying RL to the base model already preserve sufficient reasoning diversity and achieve similar improvements?**
>
> Table 4 empirically investigates this aspect. We observe that applying GRPO directly to the base Qwen3-8B leads to a performance degradation of 6.2% (33.0% →26.8%), despite the presence of sufficient reward variance for optimization.
> We argue that this failure stems from the fundamental difference between the two stages:
>
> 1. RL acts as a selector. It effectively reinforces existing behavioral patterns but struggles to introduce structurally new reasoning templates (e.g., task decomposition) from scratch.
>
> 2. SFT acts as an initializer. Self-distillation injects the decomposition prior into the model's repertoire.
>
> Thus, SFT creates the necessary representation foundation, making the subsequent RL optimization meaningful. The two stages are complementary, not redundant.
>
> These discussions will be added to the revised version of the manuscript. We are grateful for the reviewer's insightful questions, which have helped us enhance the quality of our work.

---

> > ### Author Rebuttal · Reviewer_JSvp · 2026-04-01
> >
> > Thanks for the detailed explanation. I have no extra questions.

---

> > > ### Author Response · Authors · 2026-04-03
> > >
> > > Dear Reviewer JSvp,
> > >
> > > Thank you very much for taking the time to carefully review our manuscript and for providing such thorough and insightful feedback. We truly appreciate your acknowledgement that all concerns have been fully resolved.
> > >
> > > We are grateful for your constructive engagement throughout the review process. Thank you again for your time and expertise.
> > >
> > > Best regards, The Authors

---

### Official Review · Reviewer_AVhM · 2026-03-13

**Soundness:** 3
**Presentation:** 3
**Significance:** 3
**Originality:** 2
**Overall Recommendation:** 4
**Confidence:** 3

**Summary:**

The paper introduces a framework to fix "Lazy reasoning" in reasoning models. Lazy reasoning occurs when a model exhibits minimal task decomposition yet excessive reflection. This makes the generation very verbose, and there is a loss of structural reasoning. The authors propose a two-stage solution, self-distillation and diversity-aware GRPO. In self-distillation, the model utilizes reference trajectories to generate high-quality sequential tool use data, and in diversity-aware GRPO, the authors use an entropy-aware advantage function to improve the model performance.

The method D-Core helps achieve good performance on two well-known datasets (BFCL and τ -bench)

**Compliance With Llm Reviewing Policy:**

Affirmed.

**Final Justification:**

Regarding the instability of the method, I am not fully convinced that this method will be stable when extended to other domains, or it will require a lot of hyperparameter tuning. For example, in Fig 6, changing the alpha reduces the performance of the method below GRPO. Nevertheless, I still see merit in the work, and the method performs better than previous baselines. Therefore, I have increased my score.

**Key Questions For Authors:**

Please refere to the above weakness section

**Limitations:**

The author does not discuss the limitation.

**Strengths And Weaknesses:**

### Strengths
1. The method utilizes self-distillation to solve the problem and reduce reliance on teacher models.
2. I think Lazy reasoning is a good contribution of this paper, as it identifies a key issue with current reasoning models, as they lack structural decomposition and rely heavily on reflection.

### Weakness
1. The method still relies on ground truth reference trajectories and a few-shot examples. However, in an agentic system, getting a ground truth reference can be difficult.
2. The entropy-based GRPO introduces several new hyperparameters, optimizing then can be difficult. For example, in fig. 6 the training reward changes significantly.
3. Comparison with other entropy-based GRPO is missing [1][2]
4. In Table 6, what is the performance on the BFCLv3 dataset?


[1]: Prabhakar, Akshara, et al. "Apigen-mt: Agentic pipeline for multi-turn data generation via simulated agent-human interplay." arXiv preprint arXiv:2504.03601 (2025).
[2]: Chen, Minghan, et al. "Seed-grpo: Semantic entropy enhanced grpo for uncertainty-aware policy optimization." arXiv preprint arXiv:2505.12346 (2025).

---

> ### Author Rebuttal · Authors · 2026-03-28
>
> We thank the reviewer for taking the time to review our paper and for acknowledging the self-distillation and discussion on Lazy Reasoning in D-CORE. Below, we provide a point-by-point response to the weaknesses raised by the reviewer.
>
> > **Weakness 1: The method still relies on ground truth reference trajectories and a few-shot examples. However, in an agentic system, getting a ground truth reference can be difficult.**
>
> D-CORE relies strictly on outcome labels rather than process supervision. The target $Y^{\ast}$ provides only the correct API call sequence, leaving the model to autonomously discover the intermediate reasoning and decomposition. Unlike open-ended tasks,  $Y^*$ in tool-use domains are highly scalable and require no human annotation. They are readily obtained through:
>
> Execution verification: Tool calls are deterministic. Candidate sequences can be automatically generated and verified against expected outputs in a sandbox environment (leveraged for rejection sampling in Eq. 15–18).
>
> Production logs: Real-world agent systems naturally accumulate valid interaction trajectories.Existing ecosystems (e.g., MCP, OpenClaw) and standard benchmarks (e.g., WildClawBench, LiveMCPBench, TOOLATHLON) provide the necessary sandbox environments and execution logs to directly support the D-CORE pipeline.
>
> We note that D-CORE is designed specifically for domains where execution-based verification is feasible. Applying this framework to domains with non-verifiable outcomes (e.g., open-ended dialogue) falls outside our current scope and would necessitate the integration of learned reward models.
>
> > **Weakness 2: The entropy-based GRPO introduces several new hyperparameters, optimizing then can be difficult. For example, in fig. 6 the training reward changes significantly.**
>
> DA-GRPO introduces minimal hyperparameter overhead: an entropy weight $\alpha$ and a cap $\delta$. We fix $\delta$=0.5 across all experiments, leaving $\alpha$ as the sole parameter to tune. Table 3 demonstrates that DA-GRPO is robust over a 40× range of $\alpha$ (0.01–0.4), consistently outperforming standard GRPO. The default setting ($\alpha$=0.1, $\delta$=0.5) generalizes to both 8B and 14B models without modification
>
> The fluctuations in Figure 6 represent active exploration rather than instability. While $\alpha = 0.1$ exhibits early-stage variance, it converges to the highest final reward—a behavior typical of maximum-entropy RL (e.g., SAC). Conversely, the smoother baseline curve reflects gradient stagnation from advantage collapse (Table 4) rather than superior stability.
>
> >**Weakness 3: Comparison with other entropy-based GRPO is missing**
>
> We thank the reviewer for the insightful suggestion. The results for ApiGen-MT correspond to the xLAM2 series reported in our paper. We have conducted additional experiments comparing our method with RL-ZVP(Zero-variance Prompts)[1], Seed-GRPO[2], and Dynamic sampling(DAPO)[3]. The results indicate that DA-GRPO consistently maintains superior performance across these baselines.
> Intriguingly, we observe that Seed-GRPO yields positive gains in tool-use RL tasks. While Seed-GRPO does not theoretically resolve the advantage collapse issue, its incorporation of semantic entropy proves effective in enhancing the quality of rollouts that retain non-zero advantages. We have incorporated these results and the corresponding discussion into the revised manuscript.
> | Method | $\tau$-bench Retail| $\tau$-bench Airline | $\tau$-bench Overall| BFCLv3 Overall| Avg|
> |-----|-----|-----|-----|-----|-----|
> | xLAM2-8B| 50.7| 34.0| 42.4| 72.0| 57.2|
> | GRPO|49.2|41.2|45.2|78.4|61.8|
> | GRPO+Dynamic sampling|48.3|38.8|43.6|77.8|60.7|
> |GRPO+RL-ZVP|45.6|42.4|44.0|78.5|61.3|
> |Seed-GRPO|49.8|41.2|45.2|78.5|61.9|
> |DA-GRPO|50.7|44.4|47.6|77.7|62.7|
>
> [1] T.-L. V. Le, M. Jeon, K. Vu, V. Lai, and E. Yang, "No Prompt Left Behind: Exploiting Zero-Variance Prompts in LLM Reinforcement Learning via Entropy-Guided Advantage Shaping," arXiv preprint arXiv:2509.21880, 2025.
>
> [2] Chen, Minghan, et al. "Seed-grpo: Semantic entropy enhanced grpo for uncertainty-aware policy optimization." arXiv preprint arXiv:2505.12346 (2025).
>
> [3] Q. Yu et al., "DAPO: An Open-Source LLM Reinforcement Learning System at Scale," arXiv preprint arXiv:2503.14476, 2025.
>
> > **Weakness 4: In Table 6, what is the performance on the BFCLv3 dataset?**
>
> We initially omitted the BFCLv3 results due to the stability of the performance metrics. However, we have now included these results in Table 6 of the revised manuscript to ensure completeness.
>
> | Ref Traj | Few Shot | Traj Type| Success Rate | $\tau$-bench|BFCLv3|
> |-----|-----|-----|-----|-----|-----|
> |✗| ✗ | ✗ | 73.8| 20.3 |68.4|
> |✓| ✗ |GT| 89.1| 33.8|68.3|
> |✓| ✓|GT|93.2| 37.1| 68.4|
> |✓|✓| PL|92.8| 29.6| 68.5|
>
> We thank the reviewers for their time and valuable input, which have helped us improve the paper significantly. These discussions will be added to the revised version of the manuscript

---

> > ### Author Rebuttal · Reviewer_AVhM · 2026-04-03
> >
> > Thanks to the authors for the response. Regarding the instability of the method, I am not fully convinced that this method will be stable when extended to other domains, or it will require a lot of hyperparameter tuning. For example, in Fig 6, changing the alpha reduces the performance of the method below GRPO. Nevertheless, I still see merit in the work, and the method performs better than previous baselines. Therefore, I have increased my score.

---

> > > ### Author Response · Authors · 2026-04-03
> > >
> > > Dear Reviewer AVhM,
> > >
> > > We sincerely thank you for your continued engagement with our work and for the thoughtful reassessment. We greatly appreciate your recognition of the merits of D-CORE and its improvements over previous baselines, and we are grateful that you have chosen to increase your score.
> > >
> > > Thank you again for your high-quality review and constructive feedback throughout this process. Your insights have helped us identify important directions for strengthening our work.
> > >
> > > Best regards, The Authors

---

### Decision · Program_Chairs · 2026-04-30

**Decision:**

Accept (regular)

**Comment:**

### Reviews and discussion

Scores after rebuttal: 5, 4, 4, 4 (originally 5, 4, 3, 3).

Reviewer DgHB (5, conf 3)
- Praised the well-motivated pipeline, theoretical guarantees, and ablations. Concerns about reference trajectory dependence and computational overhead were fully resolved. No follow-up questions.

Reviewer JSvp (4, conf 5)
-  Asked whether DA-GRPO guarantees convergence, whether PPO would avoid the collapse issue, and whether self-distillation is necessary. Authors provided detailed responses: DA-GRPO uses entropy as a conditional, bounded, self-terminating intervention (not a permanent objective change). GRPO is preferred over PPO for memory efficiency and stability in LRM training, and direct RL without SFT degrades performance by 6.2%. All concerns fully resolved.

Reviewer Qmbf (3 raised to 4, conf 4):
- Originally scored Weak Reject with concerns about lazy reasoning thresholds, simpler DA-GRPO alternatives, and reference trajectory sensitivity. Authors provided temperature ablation (2.0 degrades due to syntactic errors), mixed-policy comparison (LUFFY-style mixing drops 0.7%), threshold robustness analysis (Kendall's W=1.0 across 6 threshold combinations, Cohen's d=6.7), and pseudo-label results (92.8% success rate without ground truth). Raised score, noting "it addresses my main concerns."

Reviewer AVhM (3 raised to 4, conf 3):
- Concerned about ground-truth reliance, hyperparameter sensitivity, and missing entropy-GRPO baselines. Authors provided comparisons against Seed-GRPO, RL-ZVP, DAPO-style dynamic sampling, and cross-architecture results on Llama3.1-8B. In Final Justification, noted remaining stability concerns but acknowledged merit: "the method performs better than previous baselines. Therefore, I have increased my score."

### Assessment

I recommend accept. All four reviewers reached positive scores, with two raising their assessments after rebuttal. The paper identifies and quantifies the Lazy Reasoning phenomenon in multi-turn tool use, proposes a self-distillation approach that avoids teacher model dependence, and introduces DA-GRPO to address advantage collapse with a bounded entropy mechanism. Each of these holds up individually.

The rebuttal provided new comparisons against 4 entropy-GRPO baselines, temperature and mixed-policy ablations, threshold robustness analysis, cross-architecture validation, and a computational cost breakdown. Every reviewer marked their concerns as fully resolved.

The main limitation is scope: experiments are on the Qwen3 family (with one Llama cross-check), and the self-distillation stage adds roughly 3x training time. These are acknowledged and do not undermine the contribution. The paper is above the acceptance threshold and offers a practical framework for improving LRM tool use.